# FloorplanQA: A Benchmark for Spatial Reasoning in LLMs using Structured Representations

Fedor Rodionov [1]  Abdelrahman Eldesokey [1]  Michael Birsak [1]
John Femiani [2]  Bernard Ghanem [1]  Peter Wonka [1]

## Abstract

We introduce FloorplanQA*, a diagnostic benchmark for evaluating spatial reasoning in large language models (LLMs). FloorplanQA is grounded in structured representations of indoor scenes (e.g., kitchens, living rooms, bedrooms, bathrooms, and others), encoded symbolically in JSON or XML layouts. The benchmark covers core spatial tasks, including distance measurement, visibility, path finding, and object placement within constrained spaces. Our results across a variety of frontier open-source and commercial LLMs reveal that while models may succeed on shallow queries, they often fail to respect physical constraints and preserve spatial coherence, though they remain mostly robust to small spatial perturbations. FloorplanQA uncovers a blind spot in today's LLMs: inconsistent reasoning about indoor layouts. We hope this benchmark inspires new work on language models that can accurately infer and manipulate spatial and geometric properties in practical settings.

## 1. Introduction

Recent progress in large language models (LLMs) has revealed strong capabilities in structured reasoning, yet spatial inference over plausible, physically feasible environments such as indoor layouts remains poorly understood. In numerous practical applications, including architectural design, assistive planning, and embodied interaction, spatial understanding is handled through structured formats such as JSON, in which objects are specified by position, size, and orientation, rather than through images. Reasoning in these contexts requires geometric inference over symbolic layouts, not pixel-level perception.

We introduce **FloorplanQA**, a benchmark to evaluate spatial reasoning in LLMs using 2D floorplans represented in structured text-based formats. Each instance consists of a symbolic (JSON-encoded) layout paired with natural language questions that require the model to reason about spatial and geometric aspects of the scene, compute distances, evaluate placement feasibility, assess visibility, and reason about spatial constraints. FloorplanQA isolates symbolic spatial reasoning over inputs that mirror the abstractions used by designers, architects, and agents operating in structured environments. In contrast to many layout benchmarks that focus on qualitative relations or output realism, FloorplanQA emphasizes geometric consistency under explicit metric, clearance, and collision constraints.

Although LLMs can increasingly be used in tool-assisted pipelines, for example, to invoke spatial solvers or generate code, this work focuses on models' *direct, unaided* reasoning capabilities. FloorplanQA is designed to probe what LLMs can infer from structured input alone, without relying on external computation or visual grounding, in order to measure their unassisted capabilities. This baseline is important because even in tool-rich systems, models benefit from some unaided spatial ability to anticipate outputs and avoid trivial errors.

Specifically, our contributions are as follows:

- We introduce a dataset of 2,000 structured 2D layouts, including 200 layouts sourced from the Habitat Synthetic Scenes Dataset (HSSD) (Khanna et al., 2023) based on real floorplans, and 600 each from synthetically generated kitchens, living rooms, and bedrooms. All are represented in JSON and paired with spatial reasoning questions.

- We provide a diverse suite of 16,000 spatial reasoning questions, eight questions per layout, covering geometric relations, placement feasibility, spatial occupancy, and navigation.

[1] King Abdullah University of Science and Technology (KAUST), Thuwal, Saudi Arabia [2] Miami University, Oxford, OH, USA. Correspondence to: Fedor Rodionov <fedor.rodionov@kaust.edu.sa>.

*Proceedings of the 43rd International Conference on Machine Learning*, Seoul, South Korea. PMLR 306, 2026. Copyright 2026 by the author(s).

*Project page: https://OldDelorean.github.io/FloorplanQA/

- We establish structured evaluation protocols and scoring metrics that enable a fine-grained diagnosis of reasoning performance by task type and error mode.

- We conduct a comparative analysis of 15 LLMs, including 7 reasoning-focused models, as well as 8 standard models, revealing consistent failure patterns in spatial inference from symbolic input.

FloorplanQA provides a benchmark of layouts, questions, and evaluation metrics for assessing spatial reasoning in language models, focusing on symbolic floorplans that integrate geometry and semantics in ways that mirror real architectural abstractions.

## 2. Related Work

Prior benchmarks have explored spatial reasoning across vision and language domains. CLEVR (Johnson et al., 2016) is a synthetic visual question answering dataset designed to test compositional reasoning, including basic spatial relations. In real-world settings, SpatialSense (Yang et al., 2019) focuses on recognizing spatial relations in images through adversarially mined examples. Benchmarks like BabyAI (Chevalier-Boisvert et al., 2018), ALFRED (Shridhar et al., 2019), and Room-to-Room (R2R) (Anderson et al., 2017) integrate spatial understanding into embodied tasks, requiring agents to follow instructions involving navigation and object manipulation in simulated environments. Recent datasets such as ScanQA (Azuma et al., 2021) and 3DSR-Bench (Ma et al., 2024a) extend spatial reasoning evaluation into 3D environments, emphasizing the need for models to comprehend and reason about spatial relationships in three dimensions.

Vision-language models have advanced spatial reasoning but often handle it qualitatively. The VQA dataset (Agrawal et al., 2015) challenges models to answer questions about images, while VL-T5 (Cho et al., 2021) unifies vision-and-language tasks via text generation. Recent work on 3D scene graphs (Armeni et al., 2019) introduces structured representations of environments, facilitating spatial reasoning. However, these approaches may miss fine-grained geometric details necessary for precise spatial inference. Efforts like SpatialVLM (Chen et al., 2024) aim to endow vision-language models with enhanced spatial reasoning capabilities, addressing some of these limitations.

Advancements in generative models have also contributed to spatial reasoning tasks. LayoutGPT (Feng et al., 2023) leverages large language models for compositional visual planning and layout generation, while Holodeck (Yang et al., 2023) enables language-guided generation of 3D embodied AI environments. Similarly, AnyHome (Wen et al., 2023) focuses on open-vocabulary generation of structured and textured 3D homes, highlighting the integration of language

and spatial understanding in generative contexts. Infinigen Indoors (Raistrick et al., 2024) offers richly rendered 3D scenes but often produces implausible object placement due to non-convergent simulated annealing. LayoutVLM (Sun et al., 2024) and FirePlace (Huang et al., 2025) improve layout generation via optimization and constraint solving, respectively. But they assess output realism, not the model's ability to infer constraints directly. In contrast, our benchmark tests symbolic reasoning without tool-assisted refinement.

Evaluations of large language models' spatial understanding have been conducted in studies like Evaluating Spatial Understanding of Large Language Models (Yamada et al., 2023), which assesses spatial reasoning through natural language descriptions of qualitative relations (e.g., "left of," "above"), without numeric coordinates or metric computation. Additionally, benchmarks such as BALROG (Paglieri et al., 2024) test agentic reasoning in game environments (NetHack, BabyAI, etc.), where spatial reasoning emerges implicitly through sequential navigation commands rather than explicit geometric inference. While these efforts reveal important limitations in high-level spatial understanding, our benchmark isolates low-level geometric reasoning over structured coordinate-based layouts, requiring models to compute distances, angles, collision-free paths, and spatial unions directly from symbolic representations. Recent 3D-LLM surveys, such as Ma et al. (2024b), cover tasks like embodied navigation and interaction, but not symbolic geometric reasoning from structured layouts.

FloorplanQA addresses this gap by providing explicit spatial representations (object coordinates and dimensions). Unlike prior benchmarks relying on raw images or commonsense spatial language, it tests precise geometric tasks: computing distances, angles, collision-free paths, and spatial unions directly from symbolic input.

## 3. Method

### 3.1. Layouts Used in This Study

We aim to create a benchmark based on floorplan datasets, with some critical requirements; first, they need to be publicly accessible, and second, they need to include furnishings (couches, tables, chairs, etc) at a minimally plausible level. These requirements matter because FloorplanQA's questions require object-level geometry (sizes, orientations, clearances), and we need a dataset that others can actually obtain, so that results are reproducible. In practice, we were unable to find a dataset that meets these requirements, as several prominent datasets, such as SUNCG (Song et al., 2016) and HouseExpo (Li et al., 2019), are not accessible due to unresolved copyright claims. Other large-scale resources—including 3D-FRONT (Fu et al., 2021), Struc-

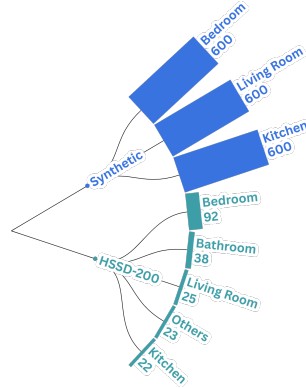

| Room Type | Style | Rectangular | L-Shaped | Open | Total |
|---|---|---|---|---|---|
| Kitchen | L-Shaped | 208 | 36 | 105 | |
| | U/G-Shaped | 42 | 128 | 2 | 600 |
| | Island-Based | 2 | 0 | 33 | |
| | Wall / Galley | 17 | 10 | 17 | |
| Living Room | Fireplace-Centric | 55 | 1 | 34 | |
| | Conversational | 66 | 3 | 37 | 600 |
| | Multi-Zone | 73 | 176 | 46 | |
| | TV-Focused | 78 | 3 | 28 | |
| Bedroom | With Workstation | 98 | 35 | 28 | |
| | Traditional | 114 | 59 | 57 | 600 |
| | Efficient Small | 65 | 3 | 8 | |
| | Welcoming Guest | 75 | 30 | 28 | |
| **Total** | | 893 | 484 | 423 | 1800 |

*Figure 1.* (Left) Illustration of the overall breakdown by room type for the entire 2,000-layout benchmark, encompassing both the synthetic component and the layouts extracted from the HSSD. (Right) Detailed distribution of the synthetic subset of the FloorplanQA benchmark (1,800 layouts) across room type, internal style, and geometric configuration.

tured3D (Zheng et al., 2019), and InteriorNet (Li et al., 2018)—are procedurally generated but impose constraints on layout diversity, furniture semantics, or downstream reuse. Datasets like CubiCasa5K (Kalervo et al., 2019) and Rent3D (Liu et al., 2015) offer fixed architectural plans from real environments but lack furnishing annotations. RPLAN (Wu et al., 2019), despite its scale, is not publicly released, and the dataset of Di et al. (2020), while large, is procedurally generated with realtor supervision and imposes restrictions on reuse. Given these legal, practical, and methodological constraints, we found a combination of data derived from HSSD Khanna et al. (2023) based on real data, extended with LLM-based synthetic data generation, to be the most viable alternative.

We incorporated 200 layouts extracted from HSSD (Khanna et al., 2023). HSSD provides 211 high-quality, human-authored 3D scenes populated with 18,656 objects across 466 semantic categories. Unlike purely procedural datasets, HSSD offers fine-grained semantics, 3D assets, and close correspondence to real interiors, making it an effective proxy for real-world interior layouts. Decorative or auxiliary objects (e.g., *vases, plants, cushions, artworks, posters, bottles, shoes, candles*) are removed to reduce clutter; see Appendix C for the full details. The original data is 3D, so we project each 3D scene to a 2D floor plan. This can result in dense polygons for each asset (repeated points, points on straight lines, excessive tokens), which would confound our ability to test 2D geometric reasoning, so we use the Douglas-Peucker algorithm (Douglas & Peucker, 1973) with $\epsilon = 0.01$m to simplify the shapes (see Figure 5 for an example, Appendix C).

To increase the diversity of our benchmark, we generated 1,800 synthetic indoor layouts using Gemini 2.5 Pro (at the time the most capable publicly available variant), with all layouts generated up-front and fixed prior to evaluation. The Gemini 2.5 Pro model has been trained on spatial reason-

ing and robotic manipulation tasks (Gemini Robotics Team et al., 2025), making it well-suited for generating geometrically plausible indoor layouts. Although our evaluation includes multiple LLMs (including Gemini variants), there is no circularity: (1) layout generation is a one-time process separate from benchmark evaluation, (2) all ground-truth answers are computed deterministically using geometric algorithms (not model outputs), and (3) correctness is verified via rule-based spatial calculations.

The generation process comprises two stages. First, we specify room geometries using explicit constraints on shape, adjacency, and design principles related to clearance, symmetry, and zoning (see Appendix B for more details). These constraints are encoded directly in the LLM prompt. Second, each room is furnished according to style-specific guidelines (e.g., a bedroom must contain a bed and storage), also defined in structured prompts, to encourage both visual realism and functional plausibility. Approximately one-third of candidate layouts are filtered out by a rule-based spatial validity filter that enforces basic clearance and accessibility constraints. The checks remove scenes with inaccessible furniture and implausible adjacencies, such as sofas blocking doors or a refrigerator overlapping a table. Full prompts, generation templates, and validation scripts are provided in the Supplementary Material.

### 3.2. Unified Dataset

Together, these two sources yield a publicly available dataset of 2,000 layouts: 1,800 synthetically generated via Gemini 2.5 Pro and 200 extracted from HSSD. The two subsets share a unified polygonal representation, enabling consistent downstream processing. Figure 2 shows examples from both sources, while Figure 1 (left) plots the room-type distribution; Figure 1 (right) reports the corresponding room/style/geometry counts for the synthetic subset.

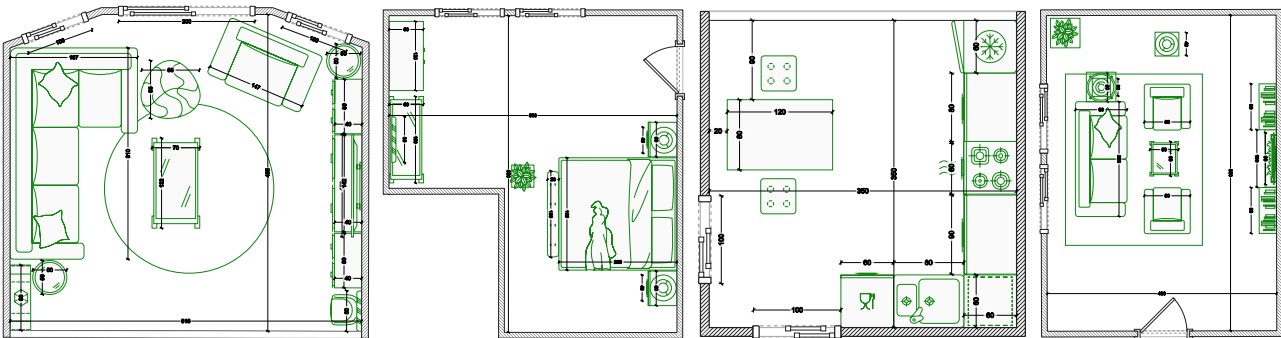

*Figure 2.* Representative layouts from FloorplanQA. **HSSD**: living room (first image); **Generated**: bedroom, kitchen and living room. Generated objects are axis-aligned boxes; HSSD uses arbitrary polygons.

Layouts are represented in a structured JSON format. Each entry contains a `layout_id`, the `room_type`, and explicit geometric descriptions. The `room_boundary` is stored as a closed polygon, while `walls` are represented separately as a list of line segments. Openings such as windows and doors are included in a dedicated `openings` field rather than flattened into the object list. All furnishings and functional elements (e.g., bed, sofa, table) are stored in the `objects` list, with each object defined by a labeled polygon. In the synthetic data, these polygons are axis-aligned bounding boxes (four points), whereas in HSSD they can exhibit arbitrary shapes and orientations. Object names are suffixed with instance identifiers (e.g., `fridge_1`, `table_3`) to ensure that referents remain unique and stable across prompt construction and answer evaluation. Coordinates are expressed in meters in a right-handed 2D Cartesian frame ($+x$ to the right, $+y$ upward), with origin at $(0, 0)$. During synthetic data generation, layouts are initially produced in a top-left origin frame (+y downward) following Gemini 2.5 Pro's native output format. All generated layouts are automatically transformed to the right-handed coordinate system before benchmark construction, ensuring consistency across synthetic and HSSD data. HSSD layouts are similarly transformed from their original 3D projection coordinates to this unified 2D frame. The prompt used to generate examples according to this schema is in Appendix K.

### 3.3. Question Taxonomy and Prompting

FloorplanQA assesses spatial reasoning by presenting models with a single natural language question per symbolic layout. Questions span a range of topological and functional types, including numeric computations (distances, areas), spatial feasibility (object placement), visibility, and requirement violations. Some questions require fine-grained metric reasoning, others test whether a model can respect physical constraints. A categorized list of question types is shown in Table 1.

We group these questions into three reasoning categories.

**Metric** tasks require explicit numerical computation, such as calculating centroids, measuring distances between objects, or evaluating the angle between an inter-object vector and a reference axis. **Topology** category involves geometric and relational reasoning, including checking placement feasibility, computing free space, or identifying whether an object blocks the direct line of sight between two others. **Dynamic** category addresses layout-changing procedures, such as repositioning an object until contact with a boundary or another object, or computing a valid collision-free path between two objects.

These categories are intended to capture the core modes of spatial reasoning in FloorplanQA, ranging from low-level geometric calculation to higher-level relational and procedural inference. While not strictly disjoint, they provide a diagnostic framework for analyzing model behavior and diagnosing failure modes.

In addition to categorizing by reasoning type, each task is also associated with an answer format code that specifies the expected output structure and the corresponding scoring rule. Scalar outputs (**N**) are scored by relative error with a default tolerance of 2%; for complex area-computation tasks (e.g., *Free Space*), the tolerance is relaxed to 5%. These tolerances are chosen to accommodate minor numerical instability in LLM outputs while remaining strict enough to distinguish correct spatial reasoning from approximation. Sensitivity studies supporting these thresholds, including tolerance sweeps showing that they affect only absolute accuracies and not model rankings, are provided in Appendix H.1. List outputs (**L**) are evaluated by set equality. Sequence outputs (**S**) must be valid (collision-free with required clearance) and are additionally scored by Fréchet distance to the reference path. We use a threshold of 0.6 m, which tolerates minor deviations in waypoint placement while rejecting qualitatively different routes (e.g., opposite sides of an obstacle). Threshold sweeps are reported in Appendix H.2.

Each question is generated by filling a parameterized tem-

*Table 1.* FloorplanQA question taxonomy. The example question shown is an instantiation of the template used to generate all questions of that type. Each task is labeled with a format code: **N** (scalar), **B** (boolean), **S** (sequence), and **L** (list), and a question reasoning category.

| Type | Example Question | Format | Category |
|------|------------------|--------|----------|
| Distance | Calculate the Euclidean distance in meters between the centroids of the fridge and the stove | N | Metric |
| Free Space | Calculate the total non-occupied floor area in square meters | N | Topology |
| View Angle | Compute the smallest absolute angle in degrees between the vector from the centroid of the sofa to the centroid of the TV and the global north vector (0, 1) | N | Metric |
| Repositioning | Calculate how far the ottoman can be moved in the left direction until it touches another object or the wall | N | Dynamic |
| Max Box | Calculate the area in square meters (m²) of the largest rectangle that can fit inside the room | N | Topology |
| Placement | Check if a 2m × 3m desk table can fit fully inside the room without overlaps | B | Topology |
| Shortest Path | Determine the shortest valid path that maintains a clearance of 15 cm from all other objects, starting from the centroid of the stove and ending at the centroid of the door | S | Dynamic |
| Visibility | Find all objects that intersect the vector from the centroid of the window to the centroid of the fireplace | L | Topology |

plate with layout-specific variables such as object names, measurements, and task-specific contextual information (e.g. units for distance, clearance for paths, or which object-types should not occlude visibility). Prompts are issued in zero-shot settings, without few-shot examples or role-based instructions. Instead, we enforce simple structural markers—such as a required checklist and a final-answer line—to encourage stepwise reasoning.

To ensure verifiable outputs, each prompt specifies a response schema consisting of a brief structured justification and a final answer line. Some models expose API-level mechanisms for structured output enforcement. We instead encode the same schema as formatting requirements in the prompt for all models to keep the prompting interface fixed across systems. For example, a distance query is phrased as:

---
**Prompt: Distance Query**

Given the layout of a {room_type} in {format},
calculate the Euclidean distance in meters between the
centroids of '{obj1}' and '{obj2}'.

---

If the format, object names, or required inputs are missing, invalid, or inconsistent, the model must return: `*Final answer*: ERROR`. Otherwise, responses must follow the scheme, for example:

---
**Prompt: Response Schema**

Begin by providing a concise checklist (3–7 bullets)
of the conceptual steps necessary for calculating
the Euclidean distance. Then, carefully walk through
each reasoning step required to calculate the distance.

Respond in the following strict format:

---

---
### Output Format
<step−by−step calculations>
∗Final answer∗: <answer>

---

This structure invokes step-by-step reasoning and each question ends with `*Final Answer*: <answer>`, enabling robust extraction even when APIs lack native structured-output support.

### 3.4. Evaluation Protocol and Scoring

Each question in FloorplanQA is paired with a reference answer computed directly from the symbolic layout, enabling fully automated and deterministic evaluation of model outputs.

**Answer extraction.** To differentiate reasoning failures from formatting issues, we apply a regex-based parsing pipeline covering expected answer patterns (e.g., '`*Final answer*`' tokens). If no answer is produced or the extracted content does not match a valid format, we count the response as an error. The proportion of invalidly formatted answers is below 1%. Per-model truncation rates and invalid-format proportions are summarized in Table 9 (Appendix F.3).

**Truncation tracking.** We track cases where no answer is returned due to token limits (API: `stop_reason = TOKEN LIMIT`), a failure mode that disproportionately affects reasoning-heavy models. We report both the raw, detailed accuracy breakdown by model, room type, and question type (Appendix F, Tables 7, 8) and the adjusted accuracy computed only over non-truncated responses (Appendix L). The adjusted scores serve as an upper bound on performance, since larger token budgets could enable more complete solutions. However, truncation occurs primarily on the most challenging questions, so valid-only accuracy

may inflate performance and should not be interpreted as a standalone metric.

### 3.5. Model Inference Setup

We evaluate both large and mid-size models, including reasoning and standard variants. Large reasoning-oriented and standard models are allocated up to 12,288 tokens per completion, enabling them to process long layout descriptions and produce multi-step outputs. Mid-size models, including the GPT family variants optimized for speed and Qwen3-30B, are limited to 8,192 tokens. These budgets reflect our observations that larger models generate longer intermediate justifications and thus consume more tokens. GPT-5 is evaluated under a distinct configuration, since its reasoning intensity cannot be disabled. We run GPT-5 with reasoning enabled and "low" verbosity and limit its output budget to 4,096 tokens, counting both reasoning and generated tokens, while accounting for its higher computational cost. GPT-5-mini is evaluated with "medium" reasoning and verbosity and an 8,192-token output budget.

For each model, we use the default inference configuration provided by its vendor, with temperature set to 0. The only exception is GPT-5 family, for which the temperature parameter is not user-configurable and defaults to 1 when reasoning is enabled.

Each model is evaluated over 1800 generated layouts and 200 layouts from HSSD, with one question from each type posed per layout. Evaluation is fully automated, from layout serialization and prompt insertion to parsing, with no manual curation.

All models are evaluated on identical input distributions and scoring criteria, enabling cross-system comparisons that are architecture-agnostic and directly comparable.

## 4. Results

We evaluate fifteen models on the full benchmark: 2,000 layouts (600 kitchens, 600 living rooms, 600 bedrooms, 200 HSSD) with eight questions each, yielding 16,000 layout–question pairs. The reasoning-oriented models include GPT-5 (OpenAI et al., 2025b), gpt-oss-120B (OpenAI et al., 2025a), DeepSeek-R1-0528 (DeepSeek-AI et al., 2025), GPT-5-mini, Gemini Flash 2.5 (Comanici et al., 2025), gpt-oss-20B, and Qwen3-30B-A3B-Thinking-2507 (Yang et al., 2025). The general-purpose models include Claude Sonnet 4 (Anthropic AI, 2025), GPT-4.1 (OpenAI et al., 2023), Moonshot Kimi-K2-Instruct (Kimi Team et al., 2025), Qwen3-Coder-480B-A35B-Instruct, Qwen3-235B-A22B-Instruct-2507, GPT-4.1-mini, Qwen3-30B-A3B-Instruct-2507, and Devstral-Small-2505 (Mistral AI et al., 2025). All models are evaluated in identical zero-shot conditions. The full taxonomy is shown in Table 1; complete per-model

results are provided in Appendix F.

### 4.1. Quantitative Results

**Room type difficulty.** Figure 3 summarizes accuracy across models and question types, with general models in the top row and reasoning models in the bottom row. Kitchens yield the highest accuracy due to simpler geometry and fewer object overlaps: on average, each kitchen layout contains 0.52 overlapping object pairs (excluding rugs), compared to 1.52–1.82 overlapping pairs in bedrooms and living rooms (Appendix D, Table 5). HSSD layouts are more complex due to non-axis-aligned polygons and substantially more overlap (4.39 overlapping pairs per layout, excluding rugs), which increases the difficulty of union, clearance, and collision reasoning. Bedrooms and living rooms fall in between. Model rankings (left panel) mainly remain consistent across synthetic and HSSD layouts, suggesting that synthetic layouts test the same spatial reasoning skills as human-authored scenes.

**Task difficulty hierarchy.** Three distinct tiers emerge from analysing the right panel of Figure 3. *Metric tasks* (*Pair Distance*, *View Angle*) achieve 75–95% averaged accuracy on synthetic layouts, where objects are axis-aligned boxes. On HSSD layouts, accuracy drops to 35–60%: models frequently fail to compute centroids of complex polygons, making computational errors when applying the shoelace formula (see Appendix J.1, J.4), compared to simple averaging for 4-point boxes. *Constraint reasoning* tasks (*Placement*, *Visibility*, *Repositioning*) yield 60–85% averaged accuracy; performance degrades when multiple geometric constraints must be jointly verified. For *Placement* and *Visibility*, baseline-guessing is ruled out by balanced ground truth (49.9% True) and low empty-list frequency (<20%), respectively (see Appendix J.5, J.7, J.2). *Optimization and planning* tasks (*Free Space*, *Max Box*, *Shortest Path*) achieve only 5–45% averaged accuracy—tasks requiring geometric unions, search, or multi-step collision-free reasoning remain largely unsolved. One exception is *Free Space* on Kitchens, yielding over 90% accuracy due to minimal object overlaps; in contrast, other layouts require computing unions of overlapping object pairs (e.g., TV and TV stand, nightstand and lamp) to correctly calculate non-occupied area (see Appendix J.3, J.8, J.6).

**Reasoning vs. general models.** Reasoning-focused models show notable improvements on geometric union tasks, gaining +10–40% on *Free Space* and *Max Box* compared to general models on generated layouts. This suggests that extended inference computation helps with complex geometric operations. However, gains on path planning remain modest. A practical limitation affects some reasoning models: Gemini Flash 2.5 and DeepSeek-R1 frequently hit token limits on larger layouts (30–50% truncation; see Table 9),

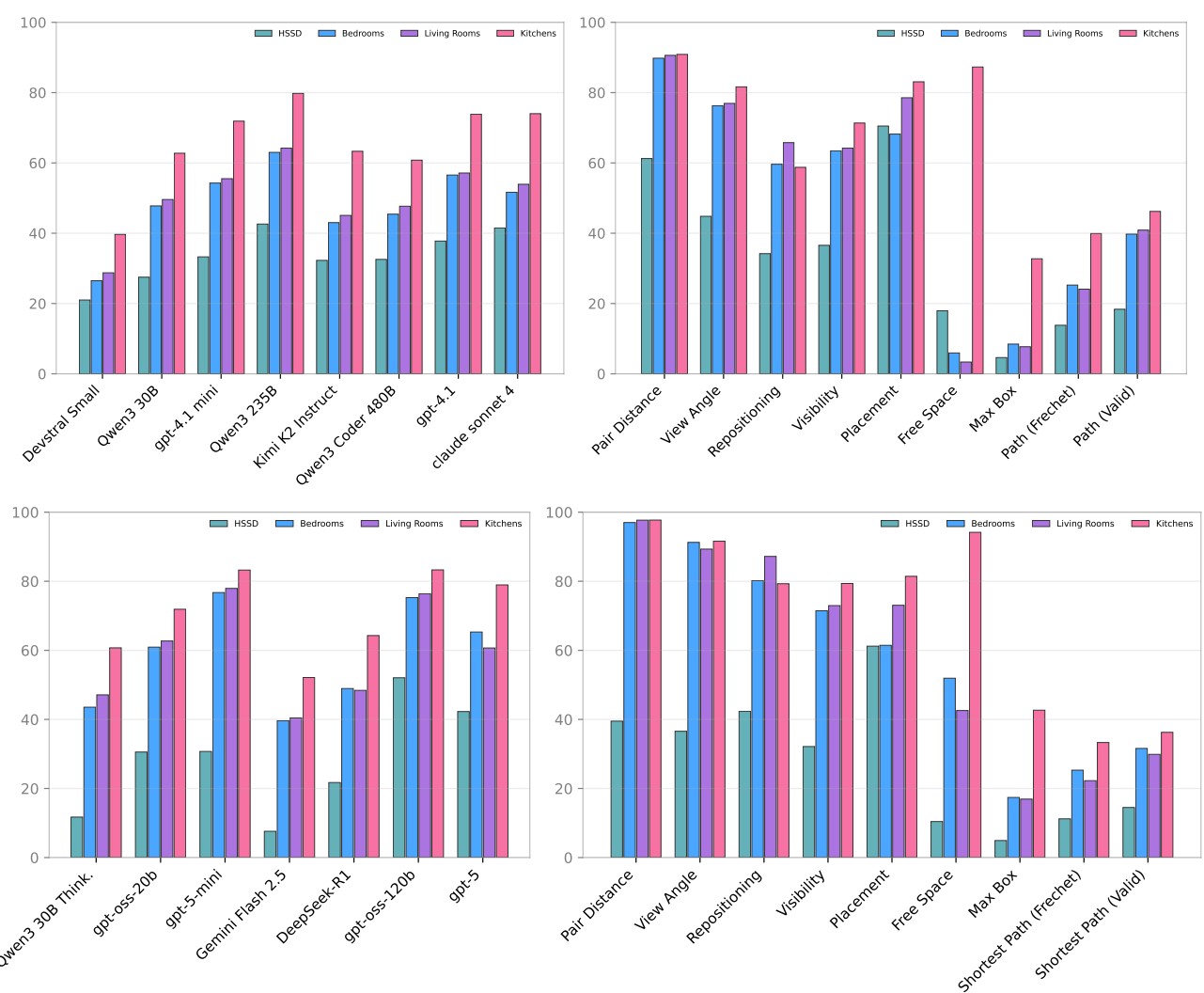

*Figure 3.* Accuracy of general (top) and reasoning (bottom) models. The left panel summarizes accuracy by model, averaged across all question types. The right panel summarizes accuracy by question, averaged across all models within the respective general or reasoning family. Each column corresponds to a specific subset in our dataset: **HSSD** layouts, plus our **Kitchens**, **Living Rooms**, and **Bedrooms**.

reducing their reported scores. Valid-only accuracy (Appendix L, Tables 16 and 18) shows these models perform competitively when responses are completed successfully.

### 4.2. Ablations

**Input format.** To test robustness to layout encoding, we replaced JSON with semantically equivalent XML for a subset of HSSD layouts across three tasks (*View Angle*, *Visibility*, *Repositioning*) and three models (gpt-oss-120B, gpt-oss-20B, Qwen3-235B). Accuracy remained stable across formats (±3 percentage points; see Appendix I.1, Table 12), suggesting models encode layout semantics independently of serialization syntax.

**Prompt and semantic variation.** We tested robustness to object reference substitution (different object names, same

template) and semantic perturbation (permuted object labels, identical geometry). Object reference substitution shows minimal effect on accuracy (Appendix I.2). Semantic perturbation reveals a task-dependent pattern: *View Angle* and *Visibility* remain stable when object labels are swapped, but *Repositioning* accuracy drops sharply (e.g., 60.5% → 40.0% for gpt-oss-120B). We hypothesize that for movement-related tasks, models rely on linguistic associations rather than geometric properties—for instance, when asked to move a "sofa" whose label has been swapped with "chair," models sometimes search for furniture that semantically resembles a sofa rather than using the object's actual geometry (Appendix I.3).

**Tool augmentation.** To isolate whether failures stem from arithmetic errors or deeper reasoning limitations, we evaluated GPT-4.1 with an integrated Python code interpreter.

Table 2 shows results on HSSD layouts. Additional results for generated layouts and GPT-4.1-mini model can be found in Appendix G, Table 10.

*Table 2.* Accuracy on HSSD layouts with and without Python code interpreter (GPT-4.1).

| Task | Raw | Tools | Δ |
|------|-----|-------|-----|
| Pair Distance | 56.0 | 99.0 | +43.0 |
| View Angle | 55.0 | 96.0 | +41.0 |
| Visibility | 46.5 | 86.5 | +40.0 |
| Repositioning | 47.0 | 83.5 | +36.5 |
| Placement | 64.5 | 95.0 | +30.5 |
| Free Space | 16.0 | 44.0 | +28.0 |
| Max Box | 7.5 | 3.0 | −4.5 |
| Shortest Path | 25.0 | 12.5 | −12.5 |

Tool augmentation substantially improves arithmetic-heavy tasks: *Pair Distance* (+43 points), *View Angle* (+41 points), *Visibility* (+40 points). However, performance on optimization and planning tasks *decreases*: *Max Box* (−4.5 points) and *Shortest Path* (−12.5 points). Inspection of failure cases (Figure 9) reveals that tool-augmented models generate Python code with incorrect algorithms—treating boundary contact as collision, or performing incomplete search over rectangle orientations. This indicates that the bottleneck for complex spatial tasks is algorithmic reasoning about spatial constraints, not numerical precision.

**Vision-language input.** We tested whether rendered floorplan images help VLMs reason about layouts. We evaluated three rendering styles: labeled bounding boxes (*Boxes*), schematic icons (*Icons*), and AI-generated photorealistic images (*GenImg*) produced from each layout with Gemini 3.1 Flash Image (*Nano Banana 2*) in image-to-image mode while keeping furniture positions fixed (Figure 4). Each was tested both alongside the JSON layout and on its own. The three rendering styles are essentially interchangeable once JSON is provided (Table 3): all three image + JSON conditions stay within ∼4 pp of the JSON-only baseline for every model, and within ∼5 pp of each other. Image-only conditions are substantially weaker, falling to 19–40%, with all three styles roughly tied. Labeled boxes are the strongest image-only modality (a +15 pp edge over icons or AI images for Gemini), suggesting that explicit object labels help more than visual realism. The structured JSON is the dominant signal: our layouts contain multiple boxes, fine-grained metric coordinates, and dense object packings that VLMs cannot reliably recover from any top-down rendering. The image does help in specific cases: Gemini gains +21 pp on *Free Space* from labeled boxes, and image + JSON modestly outperforms JSON alone on path- and angle-reasoning tasks (*Pair Distance*, *View Angle*, *Shortest Path*) for GPT-5-mini. However, these gains are task-specific and never compensate for the performance drop when JSON is removed. The per-question breakdown

(Appendix G, Table 11) reports JSON-only alongside each image condition for direct comparison.

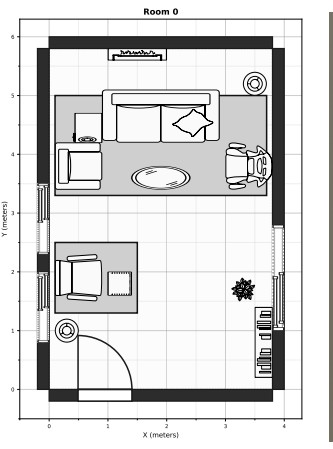 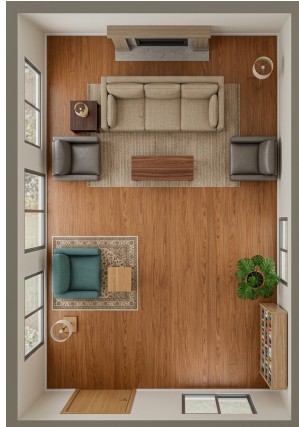

*(a)* Schematic icons          *(b)* Generated

*Figure 4.* Two rendering styles of the same Living Room layout: schematic icons used in the original benchmark (left) versus a photorealistic top-down image produced by Gemini 3.1 Flash Image in image-to-image mode (right).

## 5. Conclusion

We introduced **FloorplanQA**, a benchmark for evaluating spatial reasoning in LLMs over symbolic 2D layouts representative of architectural and planning contexts. We evaluated 15 language models (8 general, 7 reasoning) on 2,000 layouts (1,800 synthetic; 200 semi–real HSSD) across 8 types of questions spanning metric queries, visibility, placement, and collision-free path estimation.

**Key findings**. Our evaluation (Section 4) reveals a consistent capability gap: while models reliably compute some geometric properties (distances, angles), they struggle with constraint satisfaction, geometric unions, and multi-step spatial planning. Three systematic failure modes emerge: (1) Models frequently behave as if object areas were independent rather than unioned (that is, they compute the sum of areas rather than the union), which aligns with low accuracy on *Free Space* and *Max Box* (Appendix J.3, J.8). (2) Performance drops on HSSD layouts with non-axis-aligned, high-vertex polygons, and qualitative analysis indicates that many Distance/Angle errors originate in centroid computation (Appendix J.1, J.4). (3) *Shortest Path* remains difficult, especially in cluttered scenes where buffering and obstacle interactions narrow corridors (Appendix J.6); sensitivity analysis shows this across Fréchet and clearance thresholds (Appendix H.2, Fig. 13).

Reasoning-focused models show notable improvements on geometric union tasks, demonstrating that extended inference computation helps with complex geometric operations. However, gains on planning tasks remain modest, suggest-

*Table 3.* VLM accuracy (%) on the Mixed dataset (200 rooms) under seven input conditions. *JSON-only* is the structured layout with no image (baseline). *Boxes/Icons/GenImg* indicate, respectively, a simple labeled-bounding-box render, the schematic icon render, and the AI-generated photorealistic render; *+JSON* indicates that the structured layout JSON is also provided alongside the image.

| Model | JSON-only | Boxes+JSON | Icons+JSON | GenImg+JSON | Boxes-only | Icons-only | GenImg-only |
|---|---|---|---|---|---|---|---|
| GPT-4.1-mini | 59.31 | 56.19 | 55.88 | 55.69 | 23.06 | 18.94 | 19.12 |
| GPT-5-mini | 81.38 | 83.81 | 83.81 | 81.75 | 31.06 | 20.06 | 24.31 |
| Gemini 3.1 FL | 66.94 | 68.38 | 64.00 | 63.81 | 39.81 | 23.25 | 25.00 |

ing that the bottleneck lies in spatial representation rather than purely computational depth.

**Extended capabilities**. We also evaluated tool augmentation and vision-language input (Appendix G). Python Code Interpreter yields strong gains on arithmetic-heavy tasks (plus over 30–40 percentage points on Distance/Angle) but limited benefit on planning tasks, indicating failures stem from spatial reasoning rather than calculation errors. Rendered floorplan images provide selective improvements on synthetic layouts but do not consistently outperform symbolic input, suggesting symbolic representations already provide strong baselines.

**Future directions**. Two complementary directions follow. *Near term*: hybridize with external geometric solvers or symbolic planning modules, including set operations (unions/differences), centroid via shoelace, clearance buffering, oriented rectangle search, and A* path planning, to compensate for models' weaknesses in collision avoidance and clearance reasoning. *Longer term*: train with explicit spatial constraints and harder distributions (irregular, overlap-heavy layouts), and include constraint-violation exemplars and geometry-aware objectives so models learn to maintain coherence under rotation, clearance, and union operations. Beyond these directions, multi-step interaction (e.g., asking an agent to verify or revise its solution using a rendered floorplan) may further improve performance.

FloorplanQA provides a standalone benchmark for measuring progress on spatial reasoning in layout-based tasks, with applications ranging from architectural design to indoor planning.

## Acknowledgments

The work is supported by funding from King Abdullah University of Science and Technology (KAUST)—Center of Excellence for Generative AI, under award number 5940, and a gift from Google.

## Impact Statement

This paper introduces FloorplanQA, a benchmark for evaluating spatial reasoning in large language models over 2D indoor layouts. Spatial reasoning is important for many emerging applications of AI, including architectural and interior design tools, embodied agents, robotics, indoor navigation, and assistive planning systems. FloorplanQA helps measure how well artificial intelligence systems handle geometry, navigate, and deal with spatial constraints in interior layouts, making it easier to track progress in spatial reasoning.

These applications may have both positive and negative societal impacts. More capable spatial reasoning systems can assist architects and designers in the early planning stages by supporting layout analysis and helping embodied agents to better understand indoor spaces. At the same time, our experiments indicate that current language models continue to make frequent errors in geometric and constraint-based reasoning. Overestimating these capabilities can lead to invalid layouts and unreliable recommendations in downstream systems. We therefore see FloorplanQA primarily as a diagnostic tool that helps to identify these limitations.

We hope FloorplanQA will contribute to future research into models that better understand physical space and geometrical constraints. In particular, FloorplanQA may help motivate approaches that combine language models with geometric solvers, symbolic planning systems, or simulation-based verification tools. We also believe that benchmarks of this kind can help to create more realistic 3D embodied environments. More broadly, we encourage future work on human-AI collaborative tools for architecture and design, where spatial constraints can be explicitly verified, and expert users are involved in the decision-making process.

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

## A. Technical Appendix and Supplementary Material for FloorplanQA

This appendix provides comprehensive technical details supporting the main paper.

Section B describes synthetic layout generation and Section C covers HSSD extraction procedures. Section D provides detailed statistics on room complexity, object counts, and overlap distributions. Section F presents the complete evaluation breakdown by model, room type, and question category, including truncation analysis. Section G presents extended experiments with tool-augmented reasoning (Python Code Interpreter) and vision-language models using rendered floorplans. Section H provides sensitivity analyses for evaluation tolerances. Section I presents ablation studies examining input format, object reference substitution, and semantic perturbation. Section J contains qualitative failure analysis with annotated examples identifying common error patterns across question types. Section K includes full prompt templates for data generation and question asking. Section L provides supplementary accuracy tables.

All artifacts (code, generation prompts, evaluation scripts, and visualization utilities) are included with the submission as supplementary material.

## B. Details on Synthetic Data Generation

Room layouts were generated using Gemini 2.5 Pro, specifically fine-tuned for spatial reasoning and bounding-box tasks. In the following sections, we describe the detailed procedure and its constraints. All generation scripts, prompts, constraint-checking code, and random seeds are included in the code for full reproducibility.

### B.1. Room Shape Generation

We generated 600 layouts for each of the kitchens, bedrooms, and living rooms. These layouts feature a range of geometries and sizes, with clearly defined room structures that incorporate windows, doors, and corner cutouts where applicable. All layouts follow specifications tailored to each room type.

Each layout falls into one of three shape categories: rectangular, L-shaped, or open. Size categories were defined individually for each room type and were incorporated into the generation prompts based on standard assumptions about typical room dimensions.

The distribution of room shapes and sizes is shown in Table 4, and example layout types are illustrated in Fig. 2.

*Table 4.* Shape and size distribution across generated layouts for each room type.

| Room Type | Shape Distribution (Rect / L-shaped / Open) | Size Categories (in m$^2$) (Small / Medium / Large) |
|---|---|---|
| Kitchen | 40% / 40% / 20% | $\leq 7$ / 7–18 / >18 |
| Bedroom | 50% / 30% / 20% | 8–12 / 12–18 / >18 |
| Living Room | 40% / 40% / 20% | 20 / 22 / 24 |

**Geometric and Structural Constraints:** The geometry of the room was procedurally varied using prompt-based guidance to produce rectangular, L-shaped, and open-plan configurations. The following structural properties were described in the prompts but not explicitly enforced during layout generation:

- L-shaped rooms were described as rectangular spaces with a square cutout in one corner. To maintain usable proportions, each leg of the L shape was suggested to be at least 1.5 m wide and deep.

- Open-plan rooms, apart from living rooms, were prompted without doors and with one whole wall removed. This was intended to vary the room's shape and simplify the layout generation process.

- Doors were described with widths between 0.8 m and 1.0 m, randomly selected. The windows were chosen from a fixed set of widths: 0.6 m, 0.75 m, 0.9 m, 1.2 m, and 1.5 m.

- The prompts included instructions on placing all elements, such as doors, windows, and cutouts, entirely within the room boundaries.

**Window Placement:** Window placement followed prompt-based guidelines aimed at supporting daylight access and layout clarity:

- The total window length was set to exceed 15 percent of the room's floor area. It was used as a simple proxy to ensure visible window openings and visual balance along the walls.

- The windows on the same wall were the same size to support visual balance. Small, isolated windows were not used.

- Long windows, over 1.5 m, were split into segments with 0.05 to 0.15 m gaps for a more modular appearance.

- The windows were not placed opposite each other or on the same wall as a door, as such arrangements are less common and were not emphasized in the prompt design.

## B.2. Furniture and Appliance Placement

In the second stage, each room was populated with furniture and appliances based on layout style and object-specific constraints. While the styles differ by room type–such as enforcing a work triangle in kitchens, orienting seating around a focal point in living rooms, or centering beds symmetrically in bedrooms–the overall placement process followed a unified set of rules:

- All floor-standing objects must be placed without overlaps, except in semantically grouped cases (e.g., lamps on tables, chairs under tables).

- Clearance zones must be preserved around doors, main pathways, and functional elements such as beds, appliances, and desks.

- Placement follows a priority order: essential furniture is placed first, followed by optional and decorative elements only if space allows.

- Major objects like fridges, ovens, and beds must be anchored to structural walls or room boundaries.

Layouts violating any of these hard constraints, due to overlap, clearance issues, or improper attachment, were automatically discarded.

## B.3. Layout Selection Criteria

Approximately one-third of the initially generated room layouts were filtered out using a set of geometric and functional constraints. These filters were designed to ensure realistic object placement, functional usability, and architectural plausibility. While these rules are based on general design principles, they are not based on any specific design standard. Instead, they are the result of iterative development, focusing specifically on addressing cases where our prompts lead to unlikely or implausible layouts. After filtering, we retained 600 valid layouts for each room type.

- **Non-overlapping objects (with exceptions)**: Each pair of objects must satisfy axis-aligned bounding box (AABB) separation constraints, unless they belong to a known exception category. For two objects $A$ and $B$ with bounding boxes $(x_1^A, y_1^A, x_2^A, y_2^A)$ and $(x_1^B, y_1^B, x_2^B, y_2^B)$, non-overlapping requires:

$$x_2^A \leq x_1^B \quad \text{or} \quad x_2^B \leq x_1^A \quad \text{or} \quad y_2^A \leq y_1^B \quad \text{or} \quad y_2^B \leq y_1^A$$

  This constraint is not enforced for the following semantically compatible object pairs: (i) `rug` with any object placed on top of it; (ii) `lamp` with `nightstand`, `desk`, or `table`; (iii) `tv` with `tv_stand`; (iv) `chair` objects with `desk` or `table`.

  These exception pairs are considered contextually collocated or hierarchically related (e.g., support/surface relationships) and are therefore allowed to overlap.

- **Non-blocking door clearance**: Doors have physical thickness and are defined by bounding boxes $(x_1^d, y_1^d, x_2^d, y_2^d)$. A clearance zone of $door\_length$ meters is required in front of the door to ensure swing space and accessibility. The position of this zone depends on which wall the door is attached to.

- **No windows on opposite walls**: We did not include layouts with windows on directly opposite walls, as such configurations are uncommon in typical residential designs.

- **Appliances against walls or cutout edges**: Large fixtures like fridges and ovens must be flush against at least one wall or cutout boundary. This is formalized by enforcing:

$$x_1 = 0 \quad \text{or} \quad x_2 = W \quad \text{or} \quad y_1 = 0 \quad \text{or} \quad y_2 = D \quad \text{or edge of cutout}$$

For rooms with cutouts, an object may align with a cutout boundary, defined as additional wall segments with known coordinates.

These constraints were iteratively selected to address common implausible layouts generated by Gemini 2.5 Pro using our prompts. They are not universal requirements for real layouts, nor a complete set of constraints, but aim to avoid frequent sources of implausibility in generated layouts.

## C. Details on HSSD layouts selection

We curate a subset of HSSD layouts to ensure clean geometry and unambiguous supervision for spatial reasoning tasks. Starting from the raw scenes, we generate a 2D floor-plan projection and apply filtering and normalization steps (e.g., geometry cleanup and category unification).

**Object filtering.** To reduce visual clutter and retain only objects essential for spatial reasoning, we exclude purely decorative or small accessory categories. The banned labels are: `accessory`, `air conditioner`, `artwork`, `blanket`, `boots`, `bottle`, `bowl`, `box`, `book`, `brush`, `candle`, `cushion`, `decor`, `dog`, `fan`, `flower`, `frame`, `guitar`, `herb`, `hook`, `jar`, `light`, `lightbulb`, `orchids`, `pendant`, `pendulum`, `pet`, some `plants`, `poster`, `pot`, `shoes`, `socket`, `switch`, `vase`, `wreath`. We also drop subcomponents that fragment footprints without changing free space (e.g., chair/table legs; bed/armchair frames), remove partial cabinet-door leaves, and consolidate multiple near-duplicate small items by keeping a single representative instance. The resulting scenes preserve the functional layout while simplifying geometry. In Figure 5, the left panel shows the layout immediately after 2D projection; the right panel shows the simplified layout after filtering and normalization.

**Projection and cleanup.** During 2D projection, we correct mislabeled *windows* and *doors*, snap nearly collinear edges, and resolve self-intersections. To smooth rectilinear artifacts and remove redundant vertices in box-like footprints, we apply Douglas-Peucker polygon simplification with tolerance $\epsilon = 0.01$ m. This preserves the polygon's overall shape while reducing token overhead. All ground-truth answers (centroids, areas, distances) are computed from this cleaned geometry.

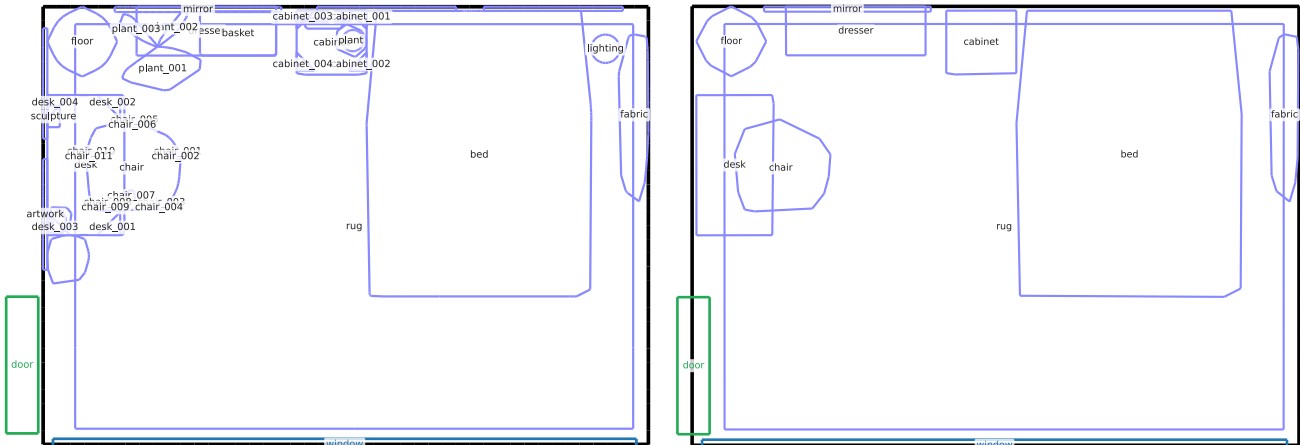

*Figure 5.* HSSD layout selection: comparison of the original (left) and simplified (right) 2D layouts. The simplified version removes redundant or overlapping objects and cleans geometry to produce well-structured input for spatial reasoning tasks.

# D. Room Statistics

Table 5 reports summary statistics for the layouts. *Overlaps* counts pairs of objects whose bounding boxes intersect. We report two versions: excluding rugs (rugs are ignored when counting overlaps) and including rugs (all overlaps counted). *Object density* is computed as the total number of objects divided by room area (objects/m²). *Vertices per layout* includes all polygon vertices from room objects.

Kitchens are typically the smallest spaces, with relatively few objects and overlaps, but a high density due to their compact geometry. Living rooms are the largest, with slightly more objects overall but lower density, reflecting their open layout. Bedrooms fall in between, with similar object counts to kitchens but more frequent overlaps.

The HSSD layouts are comparable in scale to bedrooms and living rooms in terms of area and object count, but differ in structure: objects are represented with detailed, non-axis-aligned polygons, resulting in a significantly higher vertex count. They also exhibit more overlaps than the generated layouts, reflecting their closer alignment with human-authored floorplans. In other respects, however, the distributions remain broadly consistent.

*Table 5.* Average layout statistics by room type.

| Metric | Kitchen | Bedroom | Living Room | HSSD |
|---|---|---|---|---|
| Avg. Area (m$^2$) | 12.00 | 17.76 | 20.75 | 17.95 |
| Avg. $N_{\text{objects}}$ | 10.35 | 10.76 | 11.69 | 12.20 |
| Avg. $N_{\text{overlaps}}$ (excl. rugs) | 0.52 | 1.82 | 1.52 | 4.39 |
| Avg. $N_{\text{overlaps}}$ (incl. rugs) | 0.52 | 6.25 | 6.2 | 6.65 |
| Avg. Object Density | 0.95 | 0.66 | 0.57 | 0.83 |
| Avg. Vertices per Layout | 41.39 | 43.03 | 46.77 | 152.29 |

To further illustrate these statistics, Figure 6 shows the distribution of object counts across all layouts. The histograms confirm the averages reported in Table 5: kitchens are concentrated at lower counts, typically around 10 objects; bedrooms and living rooms exhibit broader distributions with slightly higher counts; and HSSD layouts overlap in range but extend to higher counts in the tail. Overall, the object distributions remain comparable across sources, with no extreme outliers.

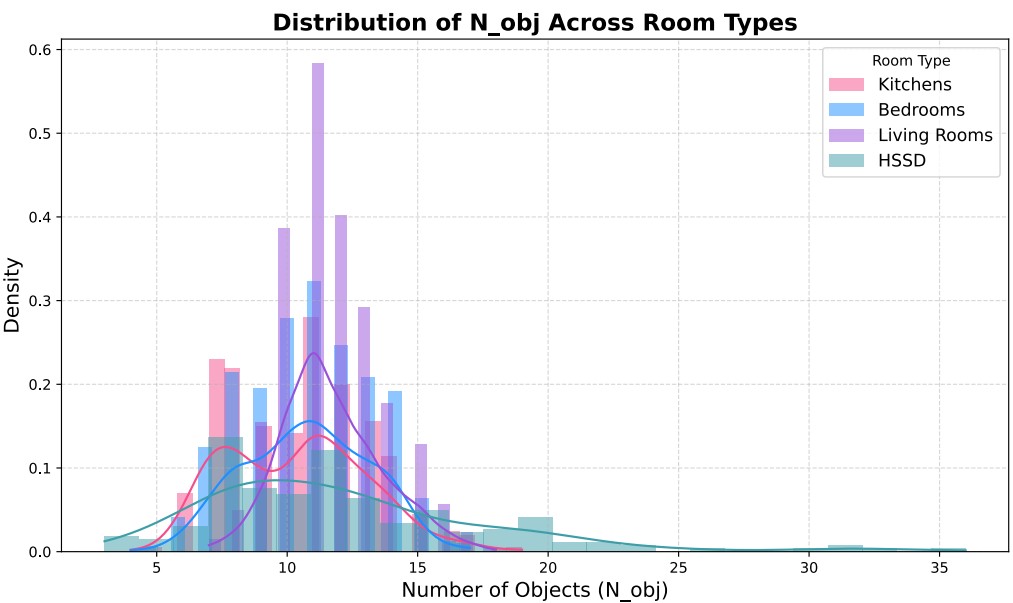

*Figure 6.* Distribution of object counts across kitchens, bedrooms, living rooms, and HSSD layouts.

# E. Layout Complexity Analysis

To identify which layout properties most strongly predict task difficulty, we computed Spearman rank correlations between per-layout properties and average accuracy. Each of the 2,000 layouts receives a single accuracy value, averaged over 15 models × 7 question types (excluding *Shortest Path*, which we analyse separately because its evaluation depends on full path validation rather than a scalar answer).

Table 6 reports the correlations. The two strongest predictors of difficulty are the number of object–object overlaps ($r = -0.517$) and the total polygon vertex count ($r = -0.451$). These capture two distinct facets of complexity: how cluttered the scene is and how geometrically detailed its objects are. Room area ($r = -0.303$) and number of objects ($r = -0.269$) are weaker but still highly significant. Object density correlates positively but modestly ($r = +0.218$), indicating that compactness alone does not restrain reasoning once overlap and vertex counts are accounted for. All reported correlations are highly significant ($p < 10^{-22}$).

*Table 6.* Spearman rank correlation between layout properties and per-layout average accuracy ($n = 2,000$ layouts; accuracy averaged across 15 models and 7 question types).

| **Property** | **Spearman** $r$ | $p$**-value** |
|---|---|---|
| $N_{\text{overlaps}}$ | $-0.517$ | $< 10^{-137}$ |
| Total polygon vertices | $-0.451$ | $< 10^{-101}$ |
| Room area (m$^2$) | $-0.303$ | $< 10^{-44}$ |
| $N_{\text{objects}}$ | $-0.269$ | $< 10^{-34}$ |
| Object density | $+0.218$ | $< 10^{-22}$ |

The effect of complexity is task-dependent. Globally scoped tasks that require aggregating geometric relations over the whole scene show the strongest sensitivity: *Free Space* ($r(N_{\text{overlaps}})=-0.588$), *Max Box* ($-0.359$), and *Visibility* ($-0.301$). Tasks defined over a single object pair are largely insensitive to global complexity: *Pair Distance* ($-0.124$) and *View Angle* ($-0.160$). $N_{\text{overlaps}}$ therefore provides a natural difficulty axis for global reasoning while leaving local-pair tasks essentially unaffected.

**Controlling difficulty.** Difficulty can be controlled by selecting layouts with more objects, overlaps, or complex polygons. The per-question-type correlations show that this control primarily affects global tasks (*Free Space*, *Max Box*) rather than local ones (*Pair Distance*, *View Angle*).

**Shortest path.** Evaluating *Shortest Path* requires geometric path validation, so we analyze it separately. HSSD paths require more waypoints than generated paths (mean 3.6 vs 2.1–2.8), and accuracy under the valid-path metric drops accordingly: 16.6% on HSSD vs 35.8–41.6% on generated layouts. This is consistent with the broader pattern – longer paths involve more planning steps, each contributing to error accumulation.

## F. Full Breakdown by Room Type, Question Type, and Model

### F.1. Detailed Accuracy by full dataset

Tables 7 and 8 report the full per-model accuracy matrix for the base and reasoning model groups, respectively. Each table covers nine question categories across four room types, resulting in a total of 36 rows. The *Shortest path* task is assessed using two complementary metrics: validity and Fréchet Distance. Together, these question types span a range of challenges, covering both reasoning-heavy and functionally grounded tasks. The base group includes eight general-purpose models, while the reasoning group includes seven models with explicit reasoning capabilities. For every model, question type accuracy is computed over a fixed set of 600 synthetic layouts (Kitchens, Living Rooms, Bedrooms) and 200 HSSD layouts, ensuring that results are directly comparable across models.

*Table 7.* Question-level accuracy on full dataset by room type for **standard models**.

| Question | Room | Claude Sonnet-4 | GPT-4.1 | Kimi-K2 Instruct | Qwen3 Coder-480B | Qwen3 235B | GPT-4.1 mini | Qwen3 30B | Devstral Small |
|---|---|---|---|---|---|---|---|---|---|
| Pair distance | K | 99.8 | 96.5 | 95.7 | 96.8 | 99.2 | 90.7 | 89.5 | 58.8 |
| | LR | 99.5 | 95.0 | 94.2 | 96.8 | 99.7 | 88.5 | 88.8 | 62.3 |
| | B | 99.7 | 96.3 | 93.3 | 96.5 | 99.5 | 87.8 | 85.2 | 60.0 |
| | HSSD | 88.0 | 56.0 | 75.5 | 66.0 | 67.0 | 37.0 | 44.5 | 56.0 |
| Placement | K | 87.8 | 78.0 | 82.2 | 80.3 | 90.2 | 86.5 | 85.5 | 74.2 |
| | LR | 80.5 | 69.0 | 73.8 | 83.2 | 89.2 | 75.2 | 85.7 | 71.8 |
| | B | 68.8 | 59.8 | 67.5 | 70.8 | 76.7 | 68.7 | 77.0 | 56.3 |
| | HSSD | 72.0 | 64.5 | 73.0 | 70.0 | 82.0 | 76.5 | 71.5 | 54.5 |
| Reposi-tioning | K | 73.8 | 63.8 | 48.7 | 45.3 | 83.5 | 66.3 | 73.7 | 14.8 |
| | LR | 79.3 | 60.3 | 56.7 | 64.5 | 91.3 | 76.8 | 72.2 | 25.0 |
| | B | 71.0 | 55.5 | 48.3 | 59.5 | 79.0 | 72.5 | 70.0 | 21.2 |
| | HSSD | 42.0 | 47.0 | 28.0 | 34.0 | 39.0 | 40.5 | 33.0 | 10.0 |
| Free space | K | 97.8 | 93.2 | 83.0 | 84.2 | 95.0 | 95.2 | 83.8 | 66.2 |
| | LR | 0.2 | 14.2 | 2.8 | 1.8 | 3.5 | 1.2 | 0.5 | 2.7 |
| | B | 2.7 | 31.2 | 1.0 | 1.3 | 8.8 | 0.8 | 1.0 | 0.7 |
| | HSSD | 35.0 | 16.0 | 24.0 | 22.0 | 17.0 | 15.5 | 7.5 | 6.5 |
| Visibility | K | 63.2 | 87.7 | 54.5 | 67.0 | 98.3 | 90.2 | 91.5 | 18.5 |
| | LR | 52.7 | 81.7 | 43.3 | 52.2 | 98.3 | 86.8 | 88.7 | 10.0 |
| | B | 57.5 | 74.8 | 41.0 | 54.0 | 96.7 | 86.3 | 86.3 | 10.8 |
| | HSSD | 20.0 | 46.5 | 22.5 | 26.5 | 70.5 | 52.0 | 45.5 | 9.0 |
| View angle | K | 92.0 | 95.3 | 69.8 | 78.8 | 97.0 | 95.8 | 74.7 | 49.7 |
| | LR | 87.7 | 93.2 | 59.7 | 75.3 | 94.0 | 93.0 | 72.2 | 40.5 |
| | B | 88.0 | 90.2 | 60.3 | 72.8 | 95.0 | 91.2 | 76.8 | 35.8 |
| | HSSD | 67.5 | 55.0 | 28.5 | 46.5 | 50.5 | 46.0 | 34.5 | 30.0 |
| Max box | K | 47.2 | 31.8 | 32.0 | 26.8 | 65.5 | 27.5 | 26.5 | 4.5 |
| | LR | 7.8 | 7.0 | 5.0 | 5.7 | 22.3 | 4.5 | 8.8 | 0.5 |
| | B | 5.8 | 6.8 | 7.3 | 5.0 | 29.2 | 4.8 | 7.0 | 1.7 |
| | HSSD | 5.0 | 7.5 | 2.5 | 2.0 | 11.5 | 4.5 | 3.0 | 1.0 |
| Shortest path (valid) | K | 59.2 | 61.7 | 52.7 | 39.7 | 45.2 | 55.3 | 21.8 | 34.2 |
| | LR | 53.0 | 51.3 | 42.8 | 37.3 | 44.8 | 47.2 | 20.3 | 30.3 |
| | B | 48.5 | 52.2 | 40.7 | 34.7 | 44.2 | 45.8 | 18.5 | 33.5 |
| | HSSD | 28.5 | 25.0 | 18.5 | 18.0 | 26.0 | 15.0 | 5.5 | 10.5 |
| Shortest path (Fréchet) | K | 45.3 | 56.8 | 51.3 | 28.2 | 44.2 | 39.5 | 17.8 | 36.2 |
| | LR | 24.7 | 42.5 | 27.3 | 12.3 | 34.7 | 26.5 | 9.2 | 15.5 |
| | B | 23.0 | 42.2 | 27.7 | 14.2 | 38.0 | 30.5 | 8.2 | 18.3 |
| | HSSD | 15.5 | 22.5 | 18.0 | 8.0 | 20.0 | 12.5 | 2.5 | 11.5 |

## F.2. Token-Limit Analysis and Valid-Only Accuracy

In addition to overall accuracy, we analyze two complementary aspects of model performance.

First, Tables 15 and 17 quantify the fraction of responses that were terminated due to the `TOKEN LIMIT` stop reason. This failure mode is particularly relevant for reasoning-oriented models, which often generate longer chain-of-thought outputs.

Second, Tables 16 and 18 report accuracy over *completed* answers only, excluding truncated outputs (e.g., token-limit terminations). This metric isolates models' reasoning performance on successfully produced, well-formed responses.

Together, these analyses complement the full accuracy tables by disentangling reasoning failures from generation truncation and formatting issues.

*Table 8.* Question-level accuracy on full dataset by room type for **reasoning models**.

| Question | Room | GPT-5 | gpt-oss 120b | DeepSeek R1-0528 | Gemini Flash 2.5 | GPT-5 mini-2025 | gpt-oss 20b | Qwen3 30B Think. |
|---|---|---|---|---|---|---|---|---|
| Pair distance | K | 99.8 | 99.3 | 98.0 | 96.3 | 100.0 | 94.2 | 97.7 |
| | LR | 98.8 | 99.3 | 99.0 | 96.0 | 99.7 | 93.5 | 97.5 |
| | B | 98.3 | 99.5 | 96.8 | 95.5 | 99.7 | 93.8 | 95.3 |
| | HSSD | 69.0 | 78.5 | 25.5 | 12.5 | 32.5 | 40.5 | 18.0 |
| Placement | K | 84.7 | 92.0 | 89.0 | 59.7 | 90.8 | 85.7 | 68.2 |
| | LR | 75.5 | 89.0 | 82.2 | 53.3 | 86.3 | 78.5 | 46.5 |
| | B | 61.2 | 83.5 | 72.0 | 35.8 | 81.2 | 62.0 | 34.5 |
| | HSSD | 70.0 | 85.0 | 79.0 | 16.0 | 75.5 | 74.5 | 28.5 |
| Reposi-tioning | K | 83.0 | 85.5 | 79.2 | 90.5 | 84.5 | 70.8 | 61.5 |
| | LR | 85.5 | 89.8 | 86.2 | 91.2 | 92.8 | 87.8 | 77.0 |
| | B | 77.8 | 83.3 | 83.3 | 85.2 | 84.3 | 78.0 | 69.2 |
| | HSSD | 49.5 | 60.5 | 47.5 | 18.5 | 53.5 | 40.0 | 27.0 |
| Free Space | K | 82.5 | 99.0 | 93.0 | 93.3 | 99.5 | 94.8 | 97.0 |
| | LR | 47.0 | 83.3 | 18.3 | 17.7 | 78.5 | 53.2 | 0.0 |
| | B | 50.5 | 87.5 | 34.8 | 33.3 | 82.2 | 74.0 | 1.2 |
| | HSSD | 19.5 | 31.0 | 6.5 | 1.0 | 5.0 | 9.0 | 1.0 |
| Visibility | K | 94.8 | 94.2 | 71.3 | 26.8 | 98.0 | 91.5 | 78.8 |
| | LR | 95.2 | 94.0 | 52.0 | 11.3 | 98.0 | 89.3 | 70.8 |
| | B | 94.2 | 92.5 | 53.5 | 11.2 | 95.5 | 89.2 | 64.2 |
| | HSSD | 57.0 | 70.0 | 10.0 | 0.5 | 39.0 | 45.5 | 3.0 |
| View Angle | K | 96.2 | 98.5 | 73.7 | 92.5 | 88.3 | 93.5 | 98.5 |
| | LR | 93.3 | 97.3 | 68.2 | 91.5 | 84.5 | 92.0 | 98.3 |
| | B | 95.2 | 98.2 | 75.2 | 93.8 | 86.8 | 91.3 | 98.3 |
| | HSSD | 59.5 | 74.0 | 13.5 | 20.0 | 25.5 | 37.5 | 26.0 |
| Max Box | K | 48.5 | 62.8 | 50.5 | 3.7 | 85.2 | 31.3 | 16.7 |
| | LR | 17.3 | 28.2 | 8.0 | 0.0 | 60.3 | 3.8 | 0.8 |
| | B | 13.0 | 30.3 | 11.0 | 0.0 | 61.2 | 5.8 | 0.5 |
| | HSSD | 5.0 | 9.5 | 2.5 | 0.0 | 17.0 | 0.5 | 0.0 |
| Shortest path (valid) | K | 64.7 | 64.2 | 12.3 | 3.2 | 52.3 | 43.0 | 14.2 |
| | LR | 21.2 | 66.0 | 12.5 | 1.5 | 53.7 | 37.7 | 16.7 |
| | B | 58.2 | 57.8 | 7.8 | 1.0 | 52.0 | 30.0 | 14.3 |
| | HSSD | 28.5 | 33.5 | 8.5 | 0.0 | 16.5 | 13.5 | 1.0 |
| Shortest path (Fréchet) | K | 56.3 | 55.5 | 11.5 | 3.2 | 50.5 | 42.2 | 14.0 |
| | LR | 12.3 | 40.5 | 9.3 | 1.3 | 47.5 | 28.5 | 16.3 |
| | B | 39.3 | 44.8 | 6.0 | 0.8 | 47.7 | 24.2 | 14.3 |
| | HSSD | 22.5 | 26.5 | 2.5 | 0.0 | 12.0 | 14.0 | 1.0 |

## F.3. Aggregated Truncation Rates

Table 9 reports aggregated truncation and error-type statistics for each model, computed by averaging outcomes across all room types and all question categories in the full dataset. A response is counted as **truncated** when it terminates with the TOKEN LIMIT stop reason. We additionally report the fraction of **invalid-format** responses that could not be parsed into a valid answer under our evaluation protocol.

For completeness, Table 9 also includes the proportions of **wrong** and **correct** responses, as well as an **alternative accuracy** (% alt), computed by extracting the last numeric value or the final list from each model's output. This alternative metric serves purely as a robustness check on the parsing and evaluation pipeline.

The right-hand columns report per-request average output and reasoning tokens (average input is ≈1,900 tokens for all models and is omitted), the total tokens consumed (input + output, in millions), and the estimated inference cost across the full benchmark (∼16,000 requests per model: 2,000 layouts × 8 question types). Reasoning tokens are reported separately where the provider exposes them (GPT-5, GPT-5-mini, Gemini 2.5 Flash); for DeepSeek-R1 and Qwen3-30B-Think they are folded into the output count. Closed-model costs use batch-tier list prices (GPT-5 $0.625/$5.00, GPT-5-mini $0.125/$1.00, GPT-4.1 $1.00/$4.00, GPT-4.1-mini $0.20/$0.80, Claude Sonnet 4 $1.50/$7.50, Gemini 2.5 Flash $0.15/$1.25 per 1M in/out, reasoning billed at output rate). Open-weights models are priced at the cheapest provider on OpenRouter, which closely matches the Nebius rates we used in practice. The total experiment cost is approximately $1,160.

*Table 9.* Error-type distribution, accuracy, and inference cost on the full dataset. Values are aggregated for each model by averaging across all room types and all questions. **Truncation** (% trunc.) quantifies the fraction of responses that were terminated due to the TOKEN LIMIT stop reason. **Invalid-format** (% invalid) denotes responses that could not be parsed into a valid answer. **Alternative accuracy** (% alt) reports accuracy computed using the last numeric value or the last list in the model's output. **AvgOut/AvgReas** are average output/reasoning tokens per request; **Total (M)** is total input + output tokens in millions; **Cost ($)** is the estimated total inference cost (see text for pricing).

| Model | % trunc. | % invalid | % wrong | % correct | % alt | AvgOut | AvgReas | Total (M) | Cost ($) |
|---|---|---|---|---|---|---|---|---|---|
| gpt-5 | 3.86 | 0.27 | 30.42 | 65.45 | 65.45 | 1,056 | 293 | 45.7 | 123.4 |
| gpt-oss-120b | 2.05 | 1.05 | 21.41 | 75.49 | 75.53 | 2,743 | — | 77.6 | 9.4 |
| DeepSeek-R1 | 31.31 | 0.78 | 17.26 | 50.65 | 50.65 | 7,473 | — | 144.9 | 269.7 |
| gpt-5-mini | 15.50 | 1.34 | 12.54 | 70.62 | 70.62 | 4,319 | 3,452 | 99.1 | 128.1 |
| Gemini Flash 2.5 | 50.58 | 0.13 | 10.14 | 39.15 | 39.15 | 1,342 | 8,323 | 44.5 | 163.1 |
| gpt-oss-20b | 16.94 | 1.18 | 21.70 | 60.18 | 60.31 | 3,860 | — | 92.8 | 9.6 |
| Qwen3-30B Think | 38.95 | 0.09 | 16.52 | 44.44 | 44.44 | 5,882 | — | 127.0 | 40.3 |
| claude-sonnet-4 | 0.05 | 0.07 | 41.09 | 58.80 | 58.80 | 866 | — | 45.0 | 150.6 |
| gpt-4.1 | 0.14 | 0.09 | 41.52 | 58.25 | 58.25 | 1,367 | — | 51.9 | 117.6 |
| Kimi-K2 | 0.15 | 0.03 | 52.04 | 47.78 | 47.78 | 685 | — | 40.9 | 42.3 |
| Qwen Coder-480B | 0.02 | 0.09 | 49.40 | 50.48 | 50.48 | 1,328 | — | 54.0 | 45.4 |
| Qwen-235B | 13.64 | 0.13 | 20.31 | 65.92 | 65.98 | 4,682 | — | 105.9 | 9.7 |
| gpt-4.1-mini | 0.67 | 0.10 | 42.18 | 57.05 | 57.05 | 1,583 | — | 55.3 | 26.3 |
| Qwen-30B Instr | 6.85 | 3.09 | 38.47 | 51.59 | 51.84 | 3,495 | — | 88.7 | 20.1 |
| Devstral Small | 2.75 | 1.56 | 65.64 | 30.05 | 30.07 | 1,101 | — | 50.8 | 8.6 |

## F.4. Radar summaries

For readability, we also provide radar visualizations that summarize accuracy and variance across models and question types, complementing the main tables.

**General models.** Figure 7 shows mean accuracy (top left) and variability (top right) across general models, as well as accuracy (bottom left) and variability (bottom right) by question type. Kitchens are consistently easier, while HSSD is the most challenging. Among models, Devstral Small performs noticeably worse, whereas Qwen3-30B achieves a level comparable to that of larger models. Across question types, *Pair Distance* and *View Angle* from the Metric category yield the highest accuracy, while more complex tasks such as *Max Box* and *Shortest Path* show lower scores and higher variance across rooms.

**Reasoning models.** Figure 8 presents analogous plots for reasoning models. GPT-family models show stronger results overall. In contrast, DeepSeek-R1 and Gemini Flash 2.5 struggle with token limits, as these models tend to produce very

long outputs according to Table 17. By question type, *Repositioning* and *Placement* are handled reliably, whereas *Max Box* and *Shortest Path* remain the most difficult as well, with high variance across rooms.

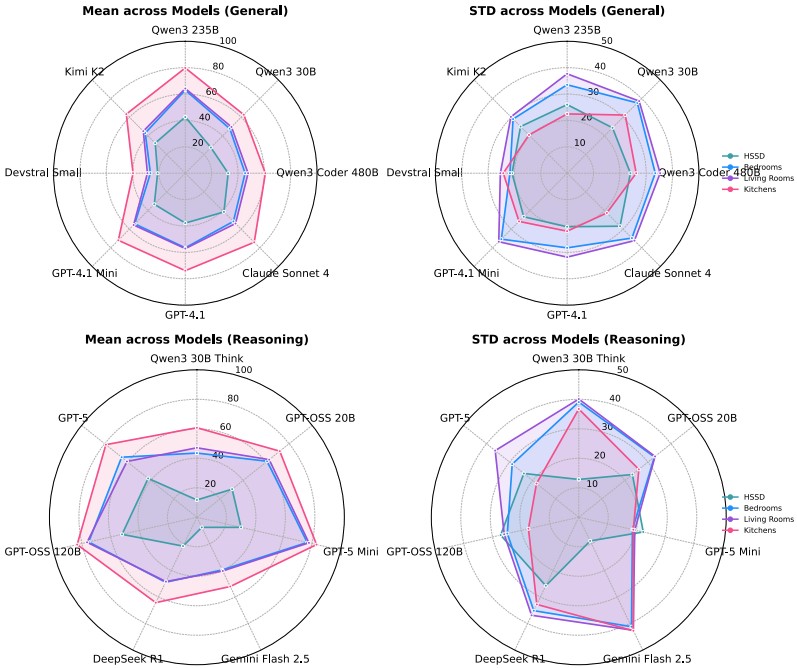

*Figure 7.* Radar plots for **general models**. Top: mean accuracy (left) and standard deviation (right) across models, grouped by room type. Bottom: mean accuracy (left) and standard deviation (right) across question types.

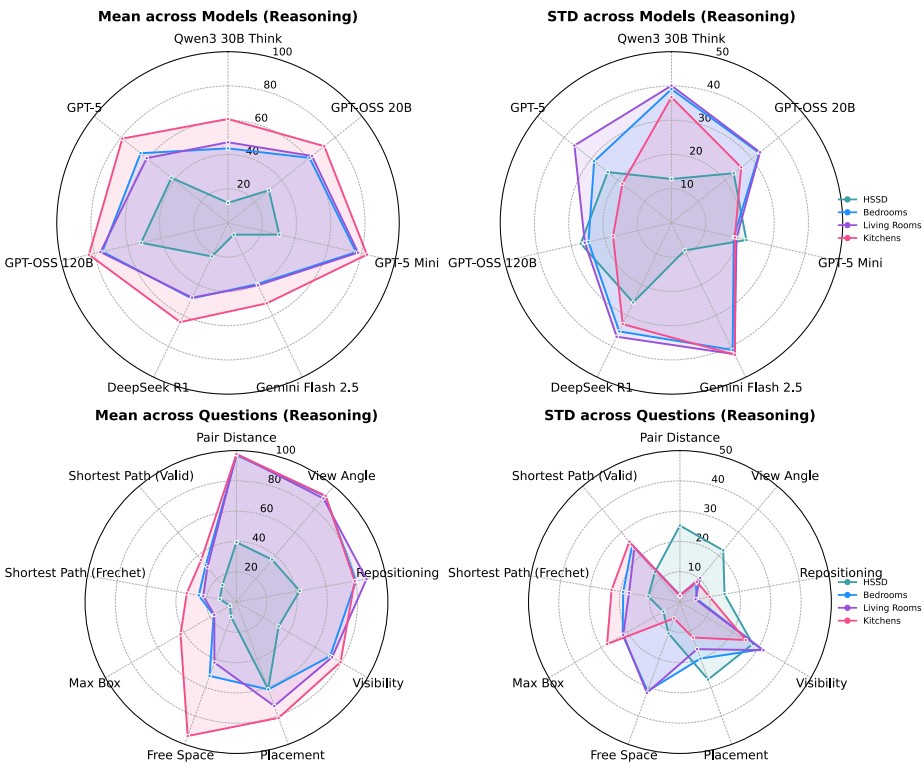

*Figure 8.* Radar plots for **reasoning models**. Top: mean accuracy (left) and standard deviation (right) across models, grouped by room type. Bottom: mean accuracy (left) and standard deviation (right) across question types.

# G. Tools and VLM experiments

To complement the text-only benchmark, we additionally evaluate (i) tool-augmented reasoning using an integrated Python interpreter and (ii) vision-language models (VLMs) using rendered floorplan images. These studies measure the potential benefit of external computation and visual input while keeping the task definitions and scoring identical to the main benchmark.

**Tool-augmented setting.** We enable the Python code interpreter tool for GPT-4.1 and GPT-4.1-mini, allowing the model to invoke a Python sandbox during inference. The model is instructed to use Python whenever numeric computation is needed (e.g., distances, angles, polygon centroids or areas), and to output a final numeric answer in the same format as the raw setting. While the interpreter improves numeric precision, models can still fail on complex tasks either due to incorrect spatial reasoning or because the generated code is incomplete or erroneous. Results are reported in Table 10.

**VLM setting and renderings.** We also evaluate VLM inputs by providing a top-down rendered floorplan image (720×720 pixels) alongside both the text question and the complete JSON layout. This multimodal setting allows models to leverage visual rendering while retaining access to precise symbolic coordinates. We use two controlled rendering styles derived from the same symbolic layout: **(i) a minimal contour-based rendering**, where room and object polygons are shown as boxes with text labels (similar to Fig. 5, right), and **(ii) an icon-based rendering**, where objects are depicted with furniture icons instead of bare contours (similar to Fig. 2). Because HSSD contains a wider variety of object categories than our current icon library, icon-based renderings are used only for synthetic layouts, while HSSD layouts use the contour style. VLM results under these input conditions are summarized in Table 10.

Table 10 shows that Python code interpreter substantially improves metric-dominated scalar tasks (distance, angle, visibility, placement), but offers limited benefit on planning-heavy tasks (Max Box, Shortest Path), indicating that remaining errors are primarily spatial rather than arithmetic. VLM inputs provide selective gains, mainly on generated layouts and for object-fit or metric cues, while not consistently improving global performance.

*Table 10.* Task accuracy for `gpt-4.1-2025-04-14` under three settings: raw text-only input (Raw), tool-augmented reasoning with a Python interpreter (Tools), and vision-language input using rendered floorplans (Img; Icons). **HSSD** columns report performance on 200 human-designed scenes. **Generated** columns report performance on 200 synthetic scenes (50 kitchens, 75 living rooms, and 75 bedrooms). Cell colors indicate gains (green) or drops (red) relative to the corresponding Raw setting for the same task.

| Question | HSSD | | | Generated | | | |
| --- | --- | --- | --- | --- | --- | --- | --- |
| | GPT-4.1 | | | GPT-4.1 | | | |
| | Raw | Tools | Img | Raw | Tools | Img | Icons |
| Pair distance | 56 | 99 | 60.5 | 95.5 | 99.5 | 99 | 99.5 |
| Placement | 64.5 | 95 | 70.5 | 65 | 92.5 | 61.5 | 71.5 |
| Repositioning | 47 | 83.5 | 44.5 | 56.5 | 48 | 42 | 39 |
| Free space | 16 | 44 | 21.5 | 41 | 28.5 | 55 | 45 |
| Visibility | 46.5 | 86.5 | 36.5 | 80 | 89 | 61.5 | 56.5 |
| View angle | 55 | 96 | 63.5 | 92 | 99 | 93.5 | 95.5 |
| Max box | 7.5 | 3 | 4 | 12 | 11.5 | 12 | 10.5 |
| Shortest path (valid) | 25 | 12.5 | 22.5 | 51 | 34 | 46 | 52 |
| Shortest path (Fréchet) | 22.5 | 12.5 | 23.5 | 40.5 | 31 | 44 | 45 |

As illustrative failure cases, we also observe that tool augmentation does not guarantee correctness for tasks that require precise geometric reasoning. In *Max Box* (Fig. 9a), the model-generated Python program performs an approximate rotated/grid search and returns a suboptimal rectangle (red, $5.57\,\mathrm{m}^2$), despite a larger valid axis-aligned solution existing (green, $7.57\,\mathrm{m}^2$). In a second example from *Repositioning* ( Fig. 9b), the ground-truth maximum downward displacement of the dishwasher is $1.96\,\mathrm{m}$, but the model's tool-based code predicts zero movement. Inspection shows that the program treats a shared boundary (the dishwasher touching another object or wall) as a collision, and therefore incorrectly concludes that no valid motion is possible for this concrete task. These cases highlight that tools reduce arithmetic imprecision, but complex tasks may still fail due to imperfect model-written algorithms and sensitivity to geometric edge cases (e.g., boundary contact vs. overlap).

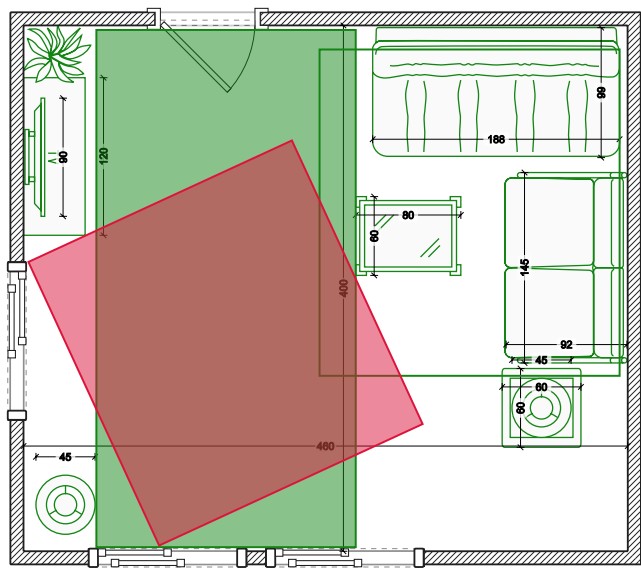
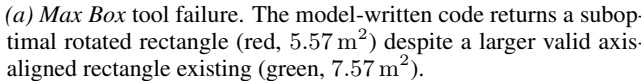
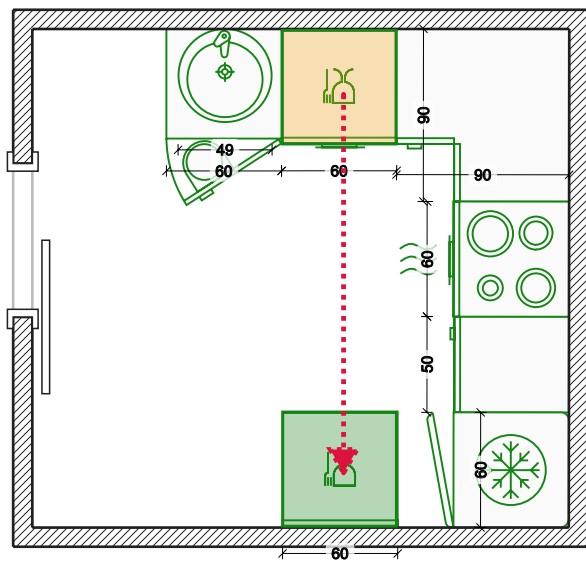

*(a) Max Box* tool failure. The model-written code returns a suboptimal rotated rectangle (red, $5.57\,\mathrm{m}^2$) despite a larger valid axis-aligned rectangle existing (green, $7.57\,\mathrm{m}^2$).

*(b) Repositioning* tool failure. The true maximum downward move is $1.96\,\mathrm{m}$, but the model's code outputs 0 by treating boundary contact as collision.

*Figure 9.* Representative failure cases in the tool-augmented setting. While the python interpreter improves numeric precision, complex tasks can still fail due to imperfect model-written code and geometric edge cases.

**Per-question vision-language results.** Table 11 reports per-question-type accuracy under all seven conditions (JSON-only baseline, three image + JSON, three image-only). Bolded cells in the image + JSON block mark $\geq 2$ pp gains over JSON-only for that model and task. On average, additional visual input helps only GPT-5-mini perform better with the Boxes and Icons styles, while the Generated style helps a little, because it can be inconsistent and may lack a clear grid structure. On the question level, two patterns stand out. First, images consistently help on *geometric/path tasks*: GPT-4.1-mini gains +8 to +10 pp on *Pair Distance* and +2 to +5 pp on *View Angle* from any image type, and GPT-5-mini gains +3 to +10 pp on *Shortest Path*—tasks where seeing the spatial layout disambiguates which JSON entries to attend to. Second, images consistently hurt on *global aggregation tasks* for GPT-4.1-mini: *Visibility* drops −15 to −17 pp and *Repositioning* drops −8 to −17 pp when images are added to JSON, suggesting the model is distracted by visual noise it cannot reconcile with the precise JSON coordinates. The most striking single-task effect is Gemini's free_space, which jumps from 46% (JSON-only) to 67% with labeled boxes but stays near baseline with icons or AI images—the text labels in the Boxes rendering evidently help this model align objects to the free-space query. Without JSON, labeled boxes are the strongest image modality across the board, but all three image-only conditions sit far below any JSON condition. Overall, image input has a task-dependent effect: it helps when resolving geometric ambiguities, but hurts when it interferes with the structured coordinate representation. In contrast, the structured JSON consistently remains the primary source of signal.

*Table 11.* Per-question accuracy (%) on the Mixed dataset (200 rooms) for the three VLMs under seven input conditions: *JSON-only* (baseline), three image + JSON conditions, and three image-only conditions. Within each model the JSON-only row is the reference; bolded image + JSON cells gain at least +2 pp over JSON-only on that task. Image-only conditions are notably weaker across the board.

| Model | Condition | Pair Dist. | Free Space | Max Box | View Angle | Visib. | Place ment | Repos. | Short. Path | Avg |
|-------|-----------|-----------|-----------|---------|-----------|--------|-----------|--------|------------|-----|
| GPT-4.1-mini | JSON-only | 88.0 | 20.0 | 8.5 | 88.5 | 87.0 | 77.0 | 57.5 | 48.0 | 59.31 |
| | Boxes+JSON | **96.0** | 19.0 | 7.0 | **93.5** | 71.0 | 73.0 | 49.0 | 41.0 | 56.19 |
| | Icons+JSON | **97.0** | 17.0 | 7.5 | **92.5** | 70.5 | 71.5 | 47.5 | 43.5 | 55.88 |
| | GenImg+JSON | **98.0** | 17.5 | 8.5 | **90.5** | 71.5 | 77.0 | 40.0 | 42.5 | 55.69 |
| | Boxes-only | 14.5 | 8.0 | 2.5 | 27.0 | 32.0 | 66.0 | 4.5 | 30.0 | 23.06 |
| | Icons-only | 10.0 | 4.5 | 4.5 | 12.5 | 11.0 | 82.0 | 6.0 | 21.0 | 18.94 |
| | GenImg-only | 5.0 | 7.0 | 6.0 | 15.5 | 17.5 | 74.5 | 5.0 | 22.5 | 19.12 |
| GPT-5-mini | JSON-only | 99.5 | 89.0 | 66.5 | 86.5 | 96.0 | 91.0 | 69.0 | 53.5 | 81.38 |
| | Boxes+JSON | 99.5 | **94.0** | 66.0 | **94.5** | 96.5 | 92.5 | **71.0** | **56.5** | **83.81** |
| | Icons+JSON | 99.5 | **93.0** | 61.5 | **92.0** | 97.0 | 92.0 | **72.0** | **63.5** | **83.81** |
| | GenImg+JSON | 99.0 | 90.5 | **69.0** | **93.5** | 97.0 | 91.5 | 61.5 | 52.0 | 81.75 |
| | Boxes-only | 19.5 | 6.0 | 6.0 | 27.5 | 51.5 | 87.0 | 7.5 | 43.5 | 31.06 |
| | Icons-only | 7.5 | 5.5 | 5.5 | 11.5 | 17.0 | 86.0 | 5.0 | 22.5 | 20.06 |
| | GenImg-only | 8.0 | 12.0 | 3.5 | 14.5 | 31.5 | 84.5 | 7.0 | 33.5 | 24.31 |
| Gemini 3.1 FL | JSON-only | 100.0 | 46.0 | 20.0 | 89.5 | 73.5 | 86.5 | 66.5 | 53.5 | 66.94 |
| | Boxes+JSON | 100.0 | **67.0** | 15.0 | 86.0 | 73.0 | 87.0 | 65.5 | 53.5 | 68.38 |
| | Icons+JSON | 100.0 | 36.0 | 17.0 | 83.0 | 73.0 | 85.5 | 62.5 | 55.0 | 64.00 |
| | GenImg+JSON | 100.0 | 33.0 | 16.0 | 87.0 | 69.5 | 85.0 | 64.0 | **56.0** | 63.81 |
| | Boxes-only | 41.0 | 22.5 | 12.0 | 35.0 | 50.0 | 83.5 | 29.0 | 45.5 | 39.81 |
| | Icons-only | 12.5 | 9.0 | 8.0 | 17.5 | 23.0 | 82.5 | 10.5 | 23.0 | 23.25 |
| | GenImg-only | 11.0 | 6.5 | 6.0 | 17.5 | 29.5 | 82.0 | 12.0 | 35.5 | 25.00 |

# H. Sensitivity studies for numeric tolerances and path thresholds

## H.1. Sensitivity studies for numeric tolerances.

We examine the robustness of our numeric evaluation tolerances for both scalar and area-based question types. For clarity and cost control, we conduct these sensitivity analyses on a representative subset of six models (three reasoning-focused and three general-purpose).

For scalar metrics (e.g., Pair distance, View angle, Repositioning, and Max Box), Figure 10 shows that relative errors are sharply concentrated near zero across models, with a clear knee before $\varepsilon_{rel} = 2\%$ (red dashed line). The accompanying tolerance sweep in Figure 12 (left) demonstrates smooth, monotone accuracy gains while changing tolerance from $0.5\%$ to $5\%$ without changing model rankings, indicating that the selection of $2\%$ lies in a stable regime rather than being outcome-sensitive.

For free-space (area) questions, Figure 11 shows a broader, heavier-tailed error distribution; nevertheless, $5\%$ tolerance lies near saturation for higher-performing models while remaining strict for weaker ones. The sweep in Figure 12 (right) confirms that relaxing area tolerance from $1\%$ to $10\%$ yields gradual improvements and preserves qualitative conclusions.

Overall, these results justify our use of a $2\%$ tolerance for scalar outputs and a $5\%$ tolerance for complex area computations.

## H.2. Sensitivity studies for path thresholds

We analyze robustness of shortest-path scoring with respect to the Fréchet threshold $\tau$. Figure 13 shows that accuracy increases smoothly as $\tau$ is relaxed, and model rankings remain almost stable across the sweep. We therefore choose $\tau = 0.6$ m as deviations within roughly $0.6$ m correspond to paths that remain traversable for a person and allow minor alternate routes without accepting qualitatively different solutions.

In addition, path validity requires collision-free traversal under a clearance buffer of $0.15$ m. This value represents a minimal safety margin; larger buffers (e.g., $0.3$ m) would incorrectly invalidate many feasible paths in compact rooms, particularly around $\sim 0.6$ m-scale kitchen utilities, and would over-penalize narrow but realistic layouts.

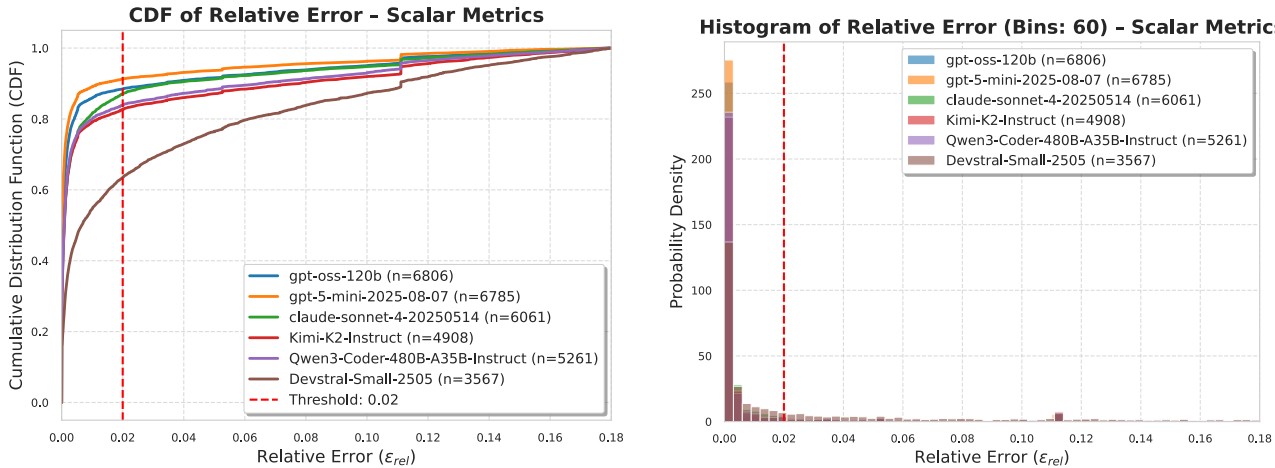

*Figure 10.* Aggregated relative-error distributions for scalar metrics across models. Left: CDF of relative error; the dashed line marks $\varepsilon_{\mathrm{rel}} = 2\%$. Right: Histogram showing a strong peak near zero and a thin long tail.

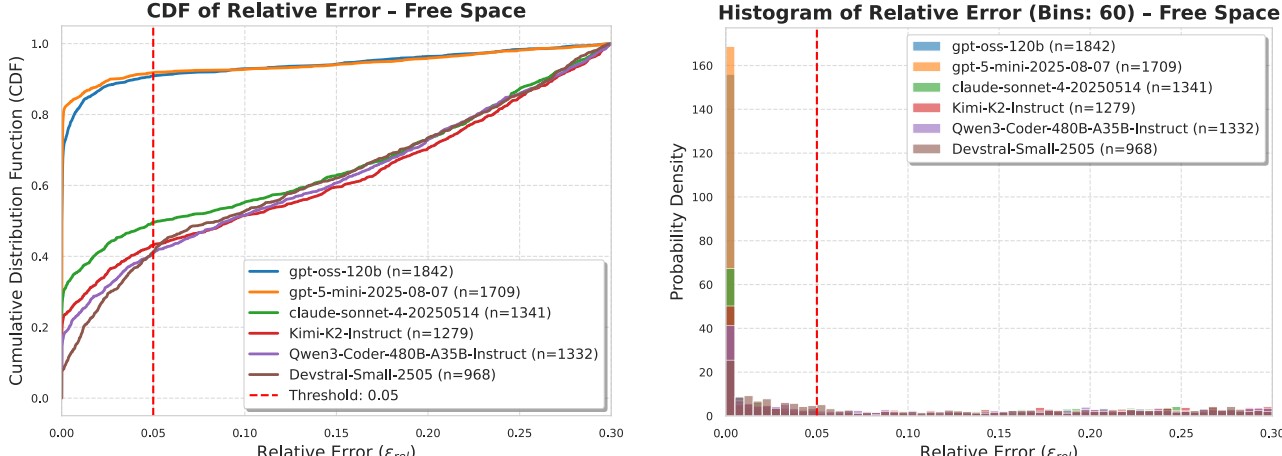

*Figure 11.* Aggregated relative-error distributions for free-space (area) questions across models. Left: CDF of relative error; the dashed line marks $\varepsilon_{\mathrm{rel}} = 5\%$. Right: Histogram showing broader, heavier-tailed errors compared to scalar tasks.

# I. Ablations

### I.1. Format Sensitivity Ablation

To evaluate the robustness of layout interpretation under alternate encodings, we replaced the standard JSON layout with a semantically equivalent XML version. This preserves all geometric and object-level content while changing only syntax. We select three tasks that show variance across HSSD layouts: *View Angle*, *Visibility*, and *Repositioning*. These span different reasoning categories (Metric, Topology, Dynamic) and have moderate baseline accuracy, where format changes are more meaningful than on near-floor tasks like *Max Box*. We test three models: two reasoning (gpt-oss-120B, gpt-oss-20B) and one general (Qwen3-235B-A22B). As shown in Table 12, accuracy stays stable across formats ($\pm 3$ points), suggesting models encode layout semantics independently of serialization syntax.

### I.2. Object Reference Substitution Ablation

To assess model robustness to object reference substitution, we conducted an ablation in which each question was regenerated using the same template but with different object references. For example, a question originally referring to a "sofa" might instead refer to a "bookshelf" in the regenerated version. This allowed us to evaluate whether model performance remains stable when object content changes while linguistic structure is preserved.

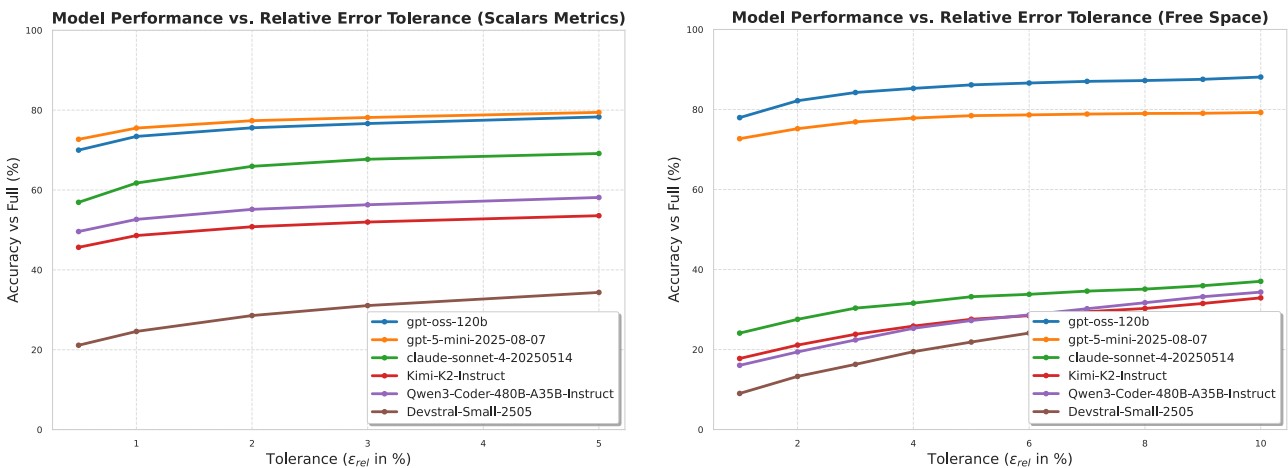

*Figure 12.* Tolerance sweeps for scalar (left) and free-space area (right) tasks. Aggregated accuracy increases smoothly as tolerance is relaxed (0.5–5% for scalars; 1–10% for areas), while model rankings remain unchanged.

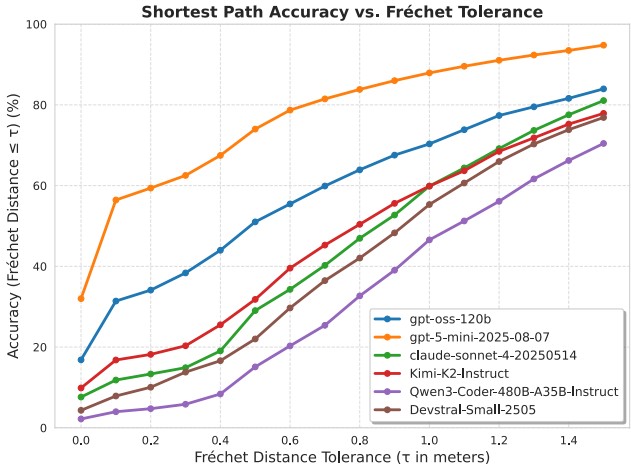

*Figure 13.* Shortest-path accuracy as a function of Fréchet tolerance $\tau$. Aggregated accuracy rises smoothly with increasing tolerance, while model ordering changes slightly. We use $\tau = 0.6$ m as a moderate, human-scale tolerance.

As shown in Table 13, accuracy under object reference substitution is broadly stable; larger models change little, and smaller ones fluctuate modestly. We observe somewhat higher sensitivity for *Repositioning*: substitutions can implicitly select different target objects or motion directions, occasionally introducing additional complexity or non-movable cases. Overall, the evaluation appears robust to object reference substitution.

### I.3. Semantic Ablation

To evaluate the model's reliance on semantic object labels rather than purely geometric reasoning, we conduct a *semantic ablation* experiment. In this setting, object identifiers in the scene are permuted (e.g., swapping `bed` and `chair` labels) while keeping all geometry unchanged. The regenerated prompts thus refer to the same physical configuration but with altered object semantics, for instance, a question originally phrased as "move the `chair` left" becomes "move the `bed` left," even though the underlying geometry is identical.

As summarized in Table 14, tasks that rely primarily on pure metric computation or topological cues, such as *View Angle* and *Visibility*, show minimal changes in accuracy, with only small fluctuations due to prompt phrasing. However, performance on the action-based *Repositioning* task drops sharply under semantic perturbation, indicating that the model partially grounds its reasoning in object semantics rather than spatial configuration alone. This suggests that large language models may

*Table 12.* Accuracy using JSON vs XML layout encoding. Each cell shows performance on the original JSON representation and its equivalent XML rendering.

| Model | Repositioning | View Angle | Visibility |
|---|---|---|---|
| gpt-oss-120B | $60.5 \rightarrow 59.0$ | $74.0 \rightarrow 74.0$ | $70.0 \rightarrow 72.0$ |
| gpt-oss-20B | $40.0 \rightarrow 39.0$ | $37.5 \rightarrow 38.5$ | $45.5 \rightarrow 43.5$ |
| Qwen3-235B-A22B | $39.0 \rightarrow 42.0$ | $50.5 \rightarrow 54.0$ | $70.5 \rightarrow 65.5$ |

*Table 13.* Accuracy under prompt variation. Each cell displays performance on the original prompt, followed by a regenerated version with alternative object references.

| Model | Repositioning | View Angle | Visibility |
|---|---|---|---|
| gpt-oss-120B | $60.5 \rightarrow 59.0$ | $74.0 \rightarrow 80.0$ | $70.0 \rightarrow 77.0$ |
| gpt-oss-20B | $40.0 \rightarrow 50.0$ | $37.5 \rightarrow 35.5$ | $45.5 \rightarrow 50.0$ |
| Qwen3-235B-A22B | $39.0 \rightarrow 49.0$ | $50.5 \rightarrow 48.0$ | $70.5 \rightarrow 74.0$ |

entangle linguistic priors with geometric inference when tasks involve physical movement or interaction.

*Table 14.* Accuracy under semantic ablation. Each cell displays performance on the original prompt, followed by a version with permuted object labels (same geometry, different semantics).

| Model | Repositioning | View Angle | Visibility |
|---|---|---|---|
| gpt-oss-120B | $60.5 \rightarrow 40.0$ | $74.0 \rightarrow 73.0$ | $70.0 \rightarrow 77.0$ |
| gpt-oss-20B | $40.0 \rightarrow 28.0$ | $37.5 \rightarrow 35.5$ | $45.5 \rightarrow 44.5$ |
| Qwen3-235B-A22B | $39.0 \rightarrow 37.0$ | $50.5 \rightarrow 43.0$ | $70.5 \rightarrow 74.0$ |

## J. Case Studies by Question Type

To better understand the sources of model failure, we conducted a qualitative analysis of representative examples from the benchmark. This section presents visualizations of selected test layouts alongside model responses. By examining both correct and incorrect outputs, we aim to identify common failure patterns and reasoning bottlenecks across different architectures.

### J.1. Pair Distance

**Task Definition**
In this task, the model is asked: *"Calculate the euclidean distance between the centroids of the `obj_1` and the `obj_2`."* In the visualization in Figure 14, these two polygons correspond to the sink and the shower; the goal is to compute their centroid-to-centroid distance.

**Ground-Truth Computation**
To establish the correct answer, we first compute the centroid $(x, y)$ of each polygon. This is done using the *shoelace formula*, which calculates the centroid based on the polygon's vertices. Once both centroids are obtained, the *Euclidean distance* between them is computed. A predicted answer is considered correct if it falls within a tolerance of $2\%$ of the ground-truth distance.

**Main Issue**
Models often fail on the HSSD dataset because they compute the centroid incorrectly. In earlier experiments, some models used the center of mass instead of the centroid; therefore, the word *centroid* is now explicitly stated in the prompt. Almost all wrong answers come from calculation mistakes in the centroid formula (areas, sums, divisions), not from the distance step. This error does not depend on polygon complexity (number of vertices).

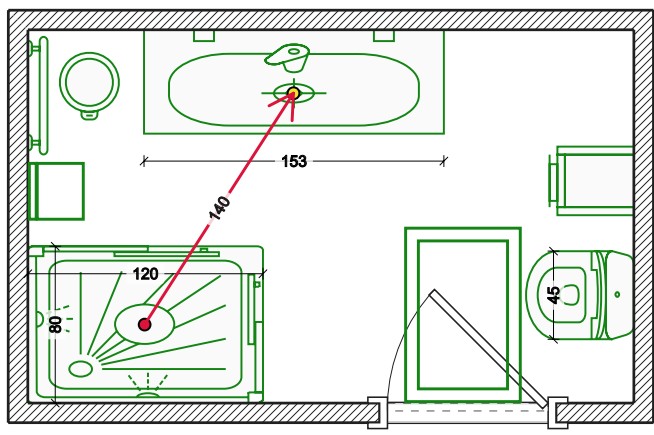
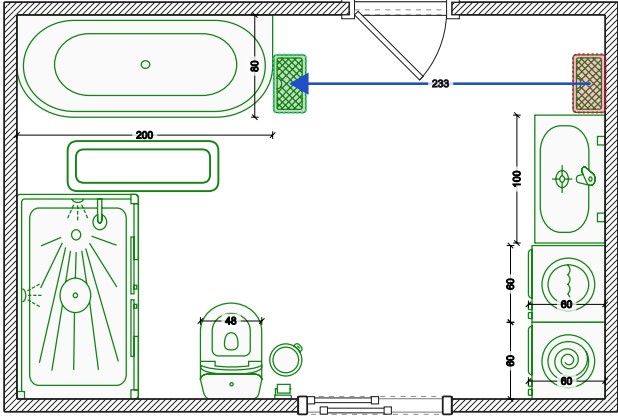

*Figure 14.* Pair Distance (bathroom): the red line indicates the centroid-to-centroid segment.

*Figure 15.* Repositioning (bedroom): red fill indicates the initial pose of `bin_2` and green fill indicates the final pose.

## J.2. Repositioning

### Task Definition

We pose the question: *"Calculate how far the `object` can be moved in the `direction` before it touches another object or the wall."* In the visualization in Figure 15, the object of interest is `bin_2` and the direction is leftward; the goal is to compute its maximum leftward translation before contacting a bathtub.

### Ground-Truth Computation

Simulate a leftward, axis-aligned slide of `bin_2`. Advance until the next step would overlap a wall or another object; take the last non-overlapping pose. Measure the travel distance from the initial position to that pose. Accept model answers within 2% tolerance.

### Main Issue

Narrow gaps and obstacles with arbitrary orientations make clearance difficult to estimate. Models often fall back to axis-aligned bounding boxes (discarding shape orientation) or omit the obstacle-union step, which leads to systematic over- or underestimation of feasible travel distances in any direction (left, right, up, or down).

## J.3. Free Space

### Task Definition

We consider the following question: *"Calculate the total non-occupied floor area in the `room`?"* In the visualization in Figure 16, the mint-highlighted area represents the space that remains free of objects within the game room; the goal is to compute its total area.

### Ground-Truth Computation

We compute the free floor area using geometric operations provided by `Shapely` (Gillies et al., 2007). All object polygons within the room are merged using `unary_union` to correctly handle overlaps. The occupied area is then obtained from this union, and the free area is computed as

$$A_{\text{free}} = A_{\text{room}} - A_{\text{union(objects)}}.$$

A model prediction is considered correct if it falls within 5% of the ground-truth free area.

### Main Issue

Accurate free-space estimation hinges on correctly handling overlapping obstacles. A common failure mode is to subtract each object's area independently, rather than forming their geometric union, which double-counts overlaps and systematically underestimates available area. For instance, when we evaluate on *HSSD* layouts using *gpt-oss-120B*, cumulative accuracy declines as object count and overlap increase (see Figure 18), consistent with this union-omission error.

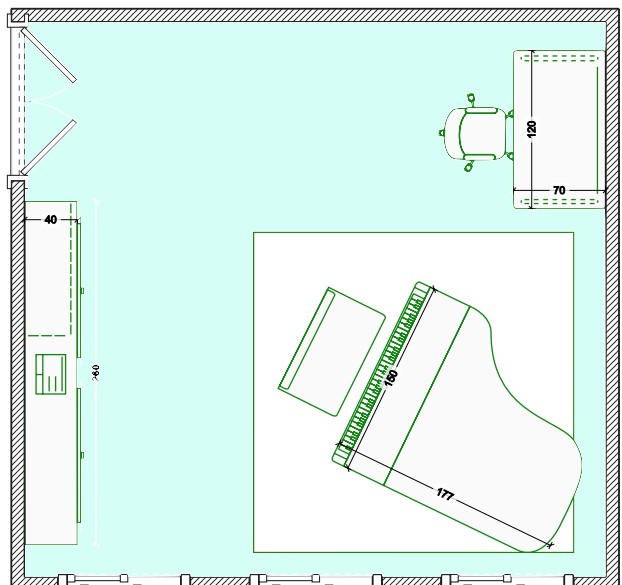

*Figure 16.* Unoccupied Floor Area (game room): the mint fill indicates the unoccupied (free-space) region within the game room.

*Figure 17.* View Angle (kitchen): smallest absolute angle between the vector from the centroid of the `chair_2` to the centroid of the `window_1` and global north $(0, 1)$.



Cumulative Accuracy ≤K

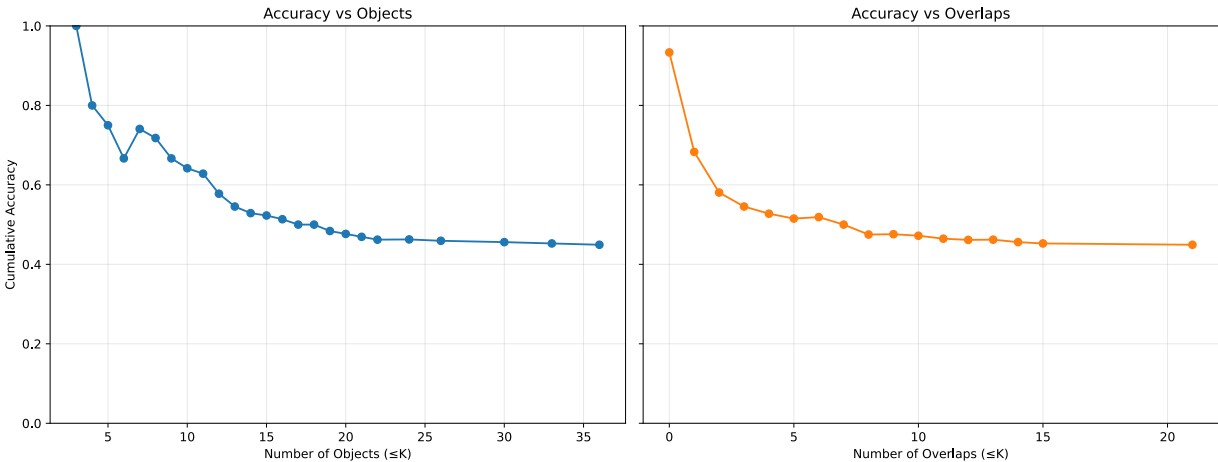

*Figure 18.* Cumulative accuracy versus layout complexity for HSSD using gpt-oss-120B on the Free-Space task. Accuracy declines sharply as object count and overlap increase, reflecting the model's difficulty in handling overlapping geometries.

## J.4. View Angle

**Task Definition**

The prompt is: *"Compute the smallest absolute angle between the vector from the centroid of* `obj_1` *to the centroid of* `obj_2` *and the global north vector* $(0, 1)$*; report θ in degrees."* In the visualization in Figure 17 (kitchen), `obj_1` is the `chair_2` and `obj_2` is the `window_1`; the goal is to return $\theta$, judged correct within a $2\%$ tolerance.

**Ground-Truth Computation**

(1) Compute centroids $\mathbf{c}_s$ (for `sofa`) and $\mathbf{c}_v$ (for `TV`) using the polygon shoelace formula.

(2) Form the displacement vector $\mathbf{d} = \mathbf{c}_v - \mathbf{c}_s$ and its unit vector $\hat{\mathbf{d}} = \dfrac{\mathbf{d}}{\|\mathbf{d}\|}$.

(3) Let the global north vector be $\mathbf{n} = (0, 1)$ (already unit length). Compute the clipped dot product: $c = \mathrm{clip}(\hat{\mathbf{d}} \cdot \mathbf{n}, -1, 1)$.

(4) Convert to degrees: $\theta = \arccos(c) \cdot \dfrac{180}{\pi} \in [0°, 180°]$.

A prediction is correct if it is within $2\%$ of the ground-truth angle $\theta$.

**Main Issue**

On HSSD layouts, most errors come from centroid *calculation* mistakes (areas/sums/divisions in the shoelace step), not from the dot product. To avoid ambiguity, the prompt explicitly says *centroid*. On synthetic layouts (4-point, axis-aligned boxes), centroids are trivial, and this issue does not appear.

## J.5. Placement

**Task Definition**

We consider the following question: *"Check if a given object can be placed in the room without overlapping walls or other objects?"* In the visualization in Figure 19, the object is a $2.5\,\mathrm{m} \times 1.0\,\mathrm{m}$ antique storage chest and the room is the living room; the goal is to determine whether a collision-free placement is possible. This task evaluates collision detection, spatial constraints, and free-space reasoning.

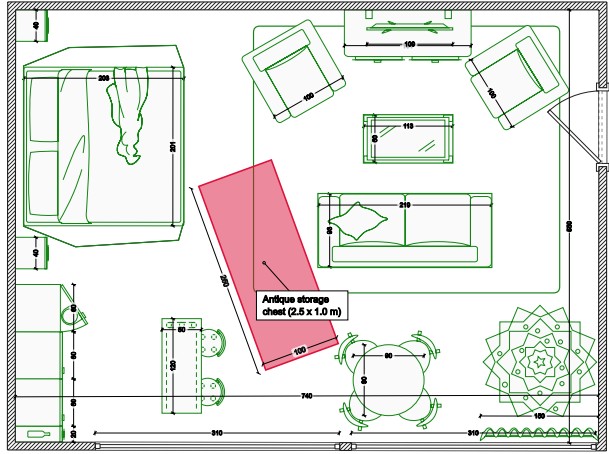

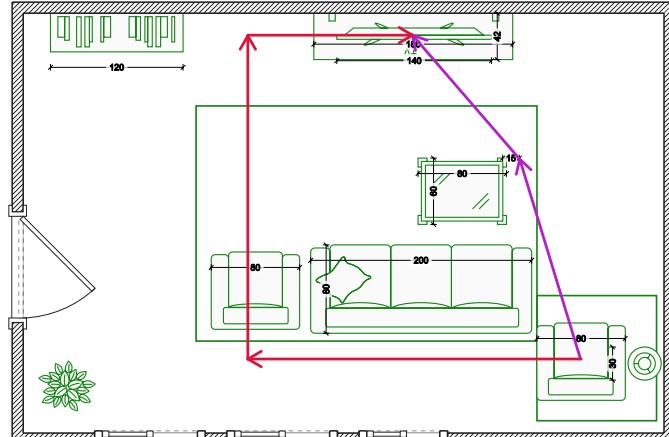

*Figure 19.* Placement (living room): determine whether an *antique storage chest* can be placed without overlapping walls or existing objects (collision-free feasibility).

*Figure 20.* Shortest Path (living room): ground-truth shortest walkable path with $15\,\mathrm{cm}$ clearance is shown in purple; the model's predicted path in red intersects armchair_1, illustrating a failure case for gpt-oss-120B due to incorrect obstacle/clearance handling.

**Ground-Truth Computation**

Form the room polygon and the union of all existing object polygons. Allow arbitrary rotation (non–axis-aligned) for the $2 \times 1.5\,\mathrm{m}$ rectangle. Search over poses: for each orientation $\theta$, test placements where the rotated rectangle is strictly inside the room polygon and has no intersection with the object union (i.e., *contains* check for the room and *disjoint* check for obstacles). If any collision-free pose exists, return True; otherwise False. Compare the model's Boolean prediction to this result.

**Main Issue**

The task is harder when non–axis-aligned placements are allowed. Models often mis-handle overlap checks under rotation and falsely report feasibility/infeasibility due to incorrect intersection computations.

## J.6. Shortest Path

**Task Definition**

The question asks: *"Determine the shortest valid path that maintains a clearance of $d$ cm from all other objects, starting from centroid of the obj_1 and ending at the centroid of the obj_2."* In Figure 20 (living room), obj_1 is the TV, obj_2 is armchair_2, and $d = 15\,\mathrm{cm}$; the goal is to compute the minimum-length collision-free path (and its length). The figure shows a failure case for gpt-oss-120B, where the predicted path violates the clearance by intersecting the armchair_1.

**Ground-Truth Computation**

Offset obstacles (equivalently, erode free space) by $0.15\,\mathrm{m}$ to enforce clearance. Run A* on the navigable grid to obtain the

shortest collision-free path polyline between `TV` and `armchair_2`. A model path is valid if it is collision-free under the same clearance; it is judged correct if its Fréchet distance to the ground-truth path is $\leq 0.6\,\mathrm{m}$.

**Main Issue**
More objects and overlaps make clearance buffering, merge obstacles, and narrow corridors, increasing failure modes. Models often mishandle overlaps, producing paths that cut through obstacles or declaring no path when one exists.

## J.7. Visibility

**Task Definition**
The prompt is: *"Find all objects that intersect the vector from the centroid of the `obj_1` to the centroid of the `obj_2`."* In the visualization in Figure 21 (office), `obj_1` is the `window` and `obj_2` is the `bin`; the goal is to return the set of objects that intersect this segment (excluding the endpoints).

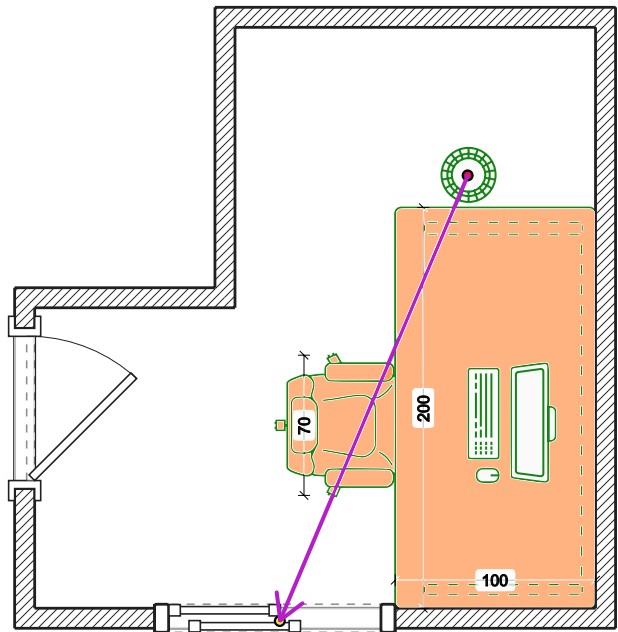

*Figure 21.* Visibility (office): the `table` and `armchair` (highlighted in orange) intersect the line segment from the centroid of the `window` to the centroid of the `bin` and constitute the correct answer.

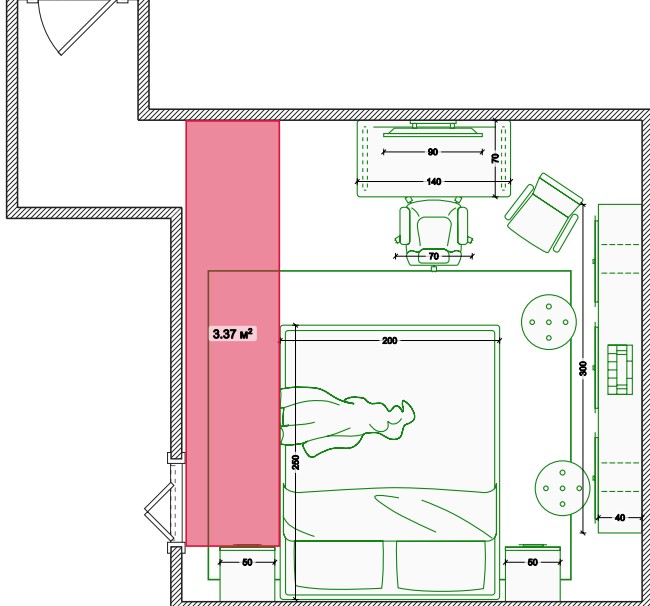

*Figure 22.* Max Box (bedroom): the red rectangle shows the largest rectangular region that can be placed without overlaps, excluding soft coverings such as rugs.

**Ground-Truth Computation**
Compute the centroids of `window` and `bin`; form the line segment between these centroids. Return the set of objects whose bounding boxes intersect this segment, excluding endpoint touches (i.e., ignore cases where the segment only touches at its endpoints).

**Main Issue**
On HSSD layouts, accuracy is slightly worse due to the larger number of objects and overlaps; the increased number of polygons along the line increases intersection ambiguity and error rates.

## J.8. Max Box

**Task Definition**
We consider the following question: *"Calculate the area in square meters of the largest rectangle that can fit inside the room"*. In the visualization in Figure 22 (bedroom), the goal is to compute the maximum-area non–axis-aligned rectangle that fits without overlapping any obstacles, and to report its area.

**Ground-Truth Computation**
Let $R$ be the room polygon and $O$ the union of all object polygons except rugs (soft coverings). Compute free space

$F = R \setminus O$. Search over orientations $\theta \in [0, \pi)$: rotate $F$ by $-\theta$, find the largest *axis-aligned* empty rectangle inside the rotated $F$, record $(w_\theta, h_\theta)$ and area $A_\theta = w_\theta h_\theta$, then map back to get $(w^*, h^*, \theta^*)$ with $A_{max} = \max_\theta A_\theta$.

### Main Issue

Harder than simple placement: the model must *optimize* size and orientation, not just answer yes/no. Allowing rotation makes the search non-convex; more objects and overlaps increase combinatorial complexity. Models often (i) ignore rotation and return an axis-aligned box, or (ii) mis-handle overlaps in free space, leading to under- or over-estimated maxima.

## K. Prompts

This section illustrates the design of prompt templates used in our benchmark. We first show a representative example of a question prompt, demonstrating how natural language templates are instantiated to elicit spatial reasoning skills (Figure 23).

Next, we present two examples of layout-generation prompts for bedrooms. The first specifies the creation of base room boundaries and openings (walls and windows) (Figure 24). The second demonstrates how furniture and objects are placed within the generated layout to yield a complete scene (Figure 25).

For completeness, full prompt templates, formatting rules, and implementation details are provided in the supplementary code to support reproducibility.

---

**Prompt: Free Space**

Given the {`room_type`} layout in {`format`}, calculate the total non-occupied (free) floor area in square meters ($m^2$).

Room layout: {`room`}

Begin with printing a concise checklist (3–7 bullets) of the conceptual steps necessary for calculating the free space. Then, carefully walk through each reasoning step required to calculate the area.

If the format, object names, or required input data are missing, invalid, or inconsistent, reply with: `*Final answer*: ERROR`

Limit your output to the step-by-step reasoning only, and do not include any internal reasoning unless explicitly requested. Clearly state the final answer on the last line using the exact format specified below.

```
### Output Format
<step-by-step calculations>
*Final answer*:  <area>
```

Where <area> is a float rounded to three decimal places, representing the free area in $m^2$. For example: `*Final answer*: 12.347`

---

*Figure 23.* Prompt for computing the largest empty rectangle area within a room layout using Chain-of-Thought reasoning.

**Prompt: Generate Bedroom Layouts**

Generate a dataset of {N} bedroom layouts in JSON format. Each layout must include:

- A unique `layout_id`

- A `room` dictionary with:
    - `width`, `depth`, `units` (meters)
    - `shape` ("rectangular", "L-shaped", or "open")
    - `shape_description`, `intended_use`, and `bed_size_suggestion`

- An `objects` list with dictionaries containing:
    - `label`, `bbox` [y0, x0, y1, x1], and a descriptive `comment`

Layouts must obey structural and spatial constraints:

- 50% rectangular, 30% L-shaped, 20% open.

- L-shape cutouts in corners; each remaining segment $\geq 1.5$ m.

- All layouts must include a door (except open types); avoid placing doors and windows on the same short wall.

- Windows must span >15% of usable floor area, with equal sizing on shared walls and valid grouping logic.

- Optional elements: fireplace (for master bedrooms), closet alcove.

- No overlap or out-of-bound placement. Fireplace must not overlap with doors/windows.

- Follow a consistent coordinate system: top-left origin, x=width (left to right), y=depth (top to bottom).

Return a JSON list of {N} valid layouts. No comments or trailing metadata.

*Figure 24.* Summarized data generation prompt for producing structured and constrained bedroom layouts.

---

**Prompt: Fill Bedroom Layout with Objects**

Given a predefined bedroom layout style and room geometry in JSON format, generate a filled 2D bird-view layout. Include a list of placed objects with their bounding boxes and explanatory comments.
Essential fields:

- Each object must have a `"label"`, `"bbox"` ([y0, x0, y1, x1]), and a descriptive `"comment"`.

- Furniture labels include: `"bed"`, `"nightstand"`, `"dresser"`, `"wardrobe"`, `"desk"`, `"chair"`, `"armchair"`, `"rug"`, `"lamp"`, etc.

- Architectural elements (`"door"`, `"window"`, `"cutout_area"`, `"fireplace"`, `"closet_alcove"`) must match the input layout and remain unmodified.

Placement priorities:

1. Place the `"bed"` according to the bed_size_suggestion and layout style.

2. Add essential storage: `"dresser"`, `"wardrobe"`, or use `"closet_alcove"` if defined.

3. Add secondary items (e.g., `"nightstand"`, `"desk"`, `"chair"`) only if space and clearance allow.

4. Add decorative or optional items (`"rug"`, `"mirror"`, `"floor_lamp"`, `"plant"`) last.

Constraints:

- Maintain at least 0.75 m clearance for walkways and door swing.

- Beds require 0.6–0.75 m of access space on sides and foot (unless against wall).

- Wardrobes/dressers need 0.6–0.8 m clearance for drawer/door use.

- No object overlap (except table lamps on nightstands or rugs under furniture).

- Use walls efficiently; avoid blocking windows unless unavoidable.

- Ensure mirror has 0.75 m clearance in front; treat `"rug"` as an anchor but optional.

Final output: a JSON list of objects, including placement and comments. No layout geometry should be altered.

*Figure 25.* Prompt for populating a bedroom layout with functionally and spatially valid object placements, following layout-specific design rules.

# L. Supplementary Accuracy Analyses

*Table 15.* % token-limit stop reason for **general models**.

| Question | Room | Claude Sonnet-4 | GPT-4.1 | Kimi-K2 Instruct | Qwen3 Coder-480B | Qwen3 235B | GPT-4.1 mini | Qwen3 30B | Devstral Small |
|---|---|---|---|---|---|---|---|---|---|
| Pair distance | K | 0.0 | 0.0 | 0.0 | 0.0 | 0.0 | 0.0 | 0.0 | 0.2 |
| | LR | 0.0 | 0.2 | 0.0 | 0.0 | 0.0 | 0.0 | 0.0 | 0.2 |
| | B | 0.0 | 0.0 | 0.2 | 0.0 | 0.0 | 0.2 | 0.0 | 0.8 |
| | HSSD | 0.0 | 0.5 | 0.0 | 0.0 | 17.5 | 8.5 | 3.5 | 1.5 |
| Placement | K | 0.0 | 0.0 | 0.0 | 0.0 | 6.5 | 0.0 | 1.2 | 1.2 |
| | LR | 0.0 | 0.0 | 0.0 | 0.0 | 6.0 | 0.0 | 1.0 | 1.5 |
| | B | 0.0 | 0.0 | 0.0 | 0.0 | 17.2 | 0.0 | 2.2 | 0.5 |
| | HSSD | 0.0 | 0.0 | 0.5 | 0.0 | 6.5 | 0.0 | 1.5 | 4.5 |
| Repositioning | K | 0.0 | 0.2 | 0.0 | 0.0 | 5.7 | 0.0 | 2.3 | 0.2 |
| | LR | 0.0 | 0.2 | 0.0 | 0.0 | 1.7 | 0.0 | 1.0 | 1.0 |
| | B | 0.0 | 0.0 | 0.0 | 0.0 | 7.5 | 0.2 | 0.8 | 1.3 |
| | HSSD | 0.0 | 0.0 | 0.0 | 0.0 | 47.0 | 2.0 | 29.0 | 4.5 |
| Free space | K | 0.0 | 0.0 | 0.2 | 0.0 | 0.2 | 0.0 | 0.3 | 2.5 |
| | LR | 0.0 | 0.3 | 0.0 | 0.0 | 4.0 | 0.0 | 1.0 | 10.2 |
| | B | 0.0 | 0.0 | 0.0 | 0.0 | 4.5 | 0.0 | 0.3 | 6.3 |
| | HSSD | 0.0 | 2.0 | 0.0 | 0.0 | 52.5 | 1.0 | 39.0 | 28.5 |
| Visibility | K | 0.2 | 0.0 | 0.0 | 0.0 | 0.0 | 0.0 | 0.2 | 0.3 |
| | LR | 0.2 | 0.0 | 0.0 | 0.0 | 0.2 | 0.2 | 0.5 | 0.3 |
| | B | 1.0 | 0.0 | 0.0 | 0.0 | 0.2 | 0.3 | 1.3 | 0.7 |
| | HSSD | 0.0 | 0.0 | 0.0 | 0.0 | 11.5 | 2.5 | 11.0 | 0.0 |
| View angle | K | 0.0 | 0.0 | 0.0 | 0.0 | 0.5 | 0.0 | 0.0 | 0.3 |
| | LR | 0.0 | 0.0 | 0.2 | 0.2 | 1.5 | 0.0 | 0.2 | 0.7 |
| | B | 0.0 | 0.0 | 0.0 | 0.3 | 0.5 | 0.0 | 0.5 | 1.2 |
| | HSSD | 0.0 | 0.5 | 0.0 | 0.0 | 26.5 | 3.0 | 17.5 | 2.5 |
| Max box | K | 0.0 | 0.0 | 0.0 | 0.0 | 22.5 | 0.0 | 6.2 | 1.5 |
| | LR | 0.0 | 0.0 | 0.7 | 0.0 | 49.8 | 0.0 | 29.3 | 0.8 |
| | B | 0.0 | 0.0 | 1.5 | 0.0 | 46.5 | 0.3 | 21.8 | 0.8 |
| | HSSD | 0.0 | 0.0 | 0.5 | 0.0 | 45.5 | 0.5 | 20.0 | 3.0 |
| Shortest path | K | 0.0 | 0.2 | 0.5 | 0.0 | 44.3 | 0.0 | 41.5 | 6.7 |
| | LR | 0.0 | 0.0 | 0.0 | 0.0 | 37.8 | 0.2 | 46.3 | 7.8 |
| | B | 0.2 | 0.0 | 0.0 | 0.0 | 40.2 | 0.0 | 49.0 | 9.2 |
| | HSSD | 0.0 | 0.0 | 0.0 | 0.0 | 35.5 | 0.0 | 43.0 | 27.0 |

*Table 16.* Question-level accuracy on completed answers for **general models**.

| Question | Room | Claude Sonnet-4 | GPT-4.1 | Kimi-K2 Instruct | Qwen3 Coder-480B | Qwen3 235B | GPT-4.1 mini | Qwen3 30B | Devstral Small |
|---|---|---|---|---|---|---|---|---|---|
| Pair distance | K | 99.8 | 96.5 | 95.7 | 96.8 | 99.2 | 90.7 | 89.5 | 58.9 |
| | LR | 99.5 | 95.2 | 94.2 | 96.8 | 99.7 | 88.5 | 88.8 | 62.4 |
| | B | 99.7 | 96.3 | 93.5 | 96.5 | 99.5 | 88.0 | 85.2 | 60.5 |
| | HSSD | 88.0 | 56.3 | 75.5 | 66.0 | 81.2 | 40.4 | 46.1 | 56.9 |
| Placement | K | 87.8 | 78.0 | 82.2 | 80.3 | 96.4 | 86.5 | 86.5 | 75.0 |
| | LR | 80.5 | 69.0 | 73.8 | 83.2 | 94.9 | 75.2 | 86.5 | 72.9 |
| | B | 68.8 | 59.8 | 67.5 | 70.8 | 92.6 | 68.7 | 78.7 | 56.6 |
| | HSSD | 72.0 | 64.5 | 73.4 | 70.0 | 87.7 | 76.5 | 72.6 | 57.1 |
| Reposi-tioning | K | 73.8 | 63.9 | 48.7 | 45.3 | 88.5 | 66.3 | 75.4 | 14.9 |
| | LR | 79.3 | 60.4 | 56.7 | 64.5 | 92.9 | 76.8 | 72.9 | 25.2 |
| | B | 71.0 | 55.5 | 48.3 | 59.5 | 85.4 | 72.6 | 70.6 | 21.4 |
| | HSSD | 42.0 | 47.0 | 28.0 | 34.0 | 73.6 | 41.3 | 46.5 | 10.5 |
| Free space | K | 97.8 | 93.2 | 83.1 | 84.2 | 95.2 | 95.2 | 84.1 | 67.9 |
| | LR | 0.2 | 14.2 | 2.8 | 1.8 | 3.6 | 1.2 | 0.5 | 3.0 |
| | B | 2.7 | 31.2 | 1.0 | 1.3 | 9.2 | 0.8 | 1.0 | 0.7 |
| | HSSD | 35.0 | 16.3 | 24.0 | 22.0 | 35.8 | 15.7 | 12.3 | 9.1 |
| Visibility | K | 63.3 | 87.7 | 54.5 | 67.0 | 98.3 | 90.2 | 91.7 | 18.6 |
| | LR | 52.8 | 81.7 | 43.3 | 52.2 | 98.5 | 87.0 | 89.1 | 10.0 |
| | B | 58.1 | 74.8 | 41.0 | 54.0 | 96.8 | 86.6 | 87.5 | 10.9 |
| | HSSD | 20.0 | 46.5 | 22.5 | 26.5 | 79.7 | 53.3 | 51.1 | 9.0 |
| View angle | K | 92.0 | 95.3 | 69.8 | 78.8 | 97.5 | 95.8 | 74.7 | 49.8 |
| | LR | 87.7 | 93.2 | 59.8 | 75.3 | 95.4 | 93.0 | 72.3 | 40.8 |
| | B | 88.0 | 90.2 | 60.3 | 72.8 | 95.5 | 91.2 | 77.2 | 36.3 |
| | HSSD | 67.5 | 55.3 | 28.5 | 46.5 | 68.7 | 47.4 | 41.8 | 30.8 |
| Max box | K | 47.2 | 31.8 | 32.2 | 26.9 | 84.5 | 27.5 | 28.2 | 4.6 |
| | LR | 7.8 | 7.0 | 5.1 | 5.7 | 44.5 | 4.5 | 12.5 | 0.5 |
| | B | 5.8 | 6.8 | 7.4 | 5.0 | 54.5 | 4.8 | 9.0 | 1.7 |
| | HSSD | 5.0 | 7.5 | 2.5 | 2.0 | 21.1 | 4.5 | 3.8 | 1.0 |
| Shortest path (valid) | K | 59.2 | 61.7 | 52.7 | 39.7 | 81.1 | 55.3 | 37.3 | 36.6 |
| | LR | 53.0 | 51.3 | 42.8 | 37.3 | 72.1 | 47.2 | 37.9 | 32.9 |
| | B | 48.6 | 52.2 | 40.7 | 34.7 | 73.8 | 45.8 | 36.3 | 36.9 |
| | HSSD | 28.5 | 25.0 | 18.5 | 18.0 | 40.3 | 15.0 | 9.7 | 14.4 |
| Shortest path (Fréchet) | K | 45.3 | 56.8 | 51.3 | 28.2 | 79.3 | 39.5 | 30.5 | 38.8 |
| | LR | 24.7 | 42.5 | 27.3 | 12.3 | 55.8 | 26.5 | 17.1 | 16.8 |
| | B | 23.0 | 42.2 | 27.7 | 14.2 | 63.5 | 30.5 | 16.0 | 20.2 |
| | HSSD | 15.5 | 22.5 | 18.0 | 8.0 | 31.0 | 12.5 | 4.4 | 15.8 |

*Table 17.* % token-limit stop reason for **reasoning models**.

| Question | Room | GPT-5 | gpt-oss 120b | DeepSeek R1-0528 | Gemini Flash 2.5 | GPT-5 mini-2025 | gpt-oss 20b | Qwen3 30B Think. |
|---|---|---|---|---|---|---|---|---|
| Pair distance | K | 0.0 | 0.0 | 1.5 | 3.3 | 0.0 | 0.7 | 2.0 |
| | LR | 0.0 | 0.0 | 0.8 | 4.0 | 0.2 | 0.7 | 2.3 |
| | B | 0.0 | 0.0 | 2.5 | 4.3 | 0.2 | 0.7 | 4.7 |
| | HSSD | 0.0 | 11.0 | 70.0 | 86.5 | 67.0 | 39.5 | 72.0 |
| Placement | K | 13.2 | 0.0 | 5.2 | 39.2 | 6.5 | 4.5 | 29.5 |
| | LR | 22.7 | 0.2 | 7.2 | 45.8 | 10.3 | 12.3 | 52.5 |
| | B | 36.2 | 0.2 | 8.2 | 63.3 | 12.8 | 25.7 | 64.5 |
| | HSSD | 27.0 | 0.0 | 6.0 | 84.0 | 16.5 | 15.0 | 70.5 |
| Reposi- tioning | K | 0.3 | 0.0 | 8.3 | 1.0 | 0.0 | 0.8 | 7.2 |
| | LR | 0.0 | 0.0 | 7.5 | 2.2 | 0.0 | 0.5 | 6.7 |
| | B | 0.2 | 0.0 | 3.2 | 1.7 | 0.0 | 0.8 | 8.3 |
| | HSSD | 2.0 | 0.5 | 33.5 | 76.0 | 28.5 | 22.5 | 63.0 |
| Free space | K | 1.2 | 0.0 | 5.5 | 2.0 | 0.0 | 0.8 | 2.3 |
| | LR | 0.2 | 0.2 | 41.7 | 41.7 | 1.7 | 13.7 | 2.3 |
| | B | 0.2 | 0.0 | 12.2 | 30.7 | 1.0 | 7.8 | 3.0 |
| | HSSD | 5.0 | 28.5 | 79.5 | 96.5 | 85.0 | 77.5 | 98.5 |
| Visibility | K | 0.0 | 1.5 | 26.7 | 72.8 | 0.0 | 0.3 | 19.2 |
| | LR | 0.0 | 1.7 | 46.8 | 88.2 | 0.3 | 1.0 | 26.3 |
| | B | 0.0 | 1.2 | 44.5 | 88.3 | 1.5 | 0.2 | 33.3 |
| | HSSD | 0.0 | 5.0 | 87.0 | 99.5 | 56.5 | 29.0 | 96.0 |
| View angle | K | 0.0 | 0.0 | 25.5 | 5.5 | 11.2 | 1.0 | 1.0 |
| | LR | 0.0 | 0.0 | 30.8 | 5.0 | 14.3 | 1.3 | 0.7 |
| | B | 0.0 | 0.0 | 23.8 | 3.7 | 12.7 | 1.2 | 0.8 |
| | HSSD | 0.0 | 7.5 | 84.5 | 75.5 | 73.0 | 38.0 | 66.0 |
| Max box | K | 0.0 | 0.0 | 30.2 | 95.7 | 2.0 | 28.8 | 62.7 |
| | LR | 0.0 | 0.0 | 63.5 | 100.0 | 5.7 | 61.0 | 98.0 |
| | B | 0.0 | 0.0 | 59.3 | 100.0 | 4.7 | 56.0 | 98.2 |
| | HSSD | 0.0 | 0.0 | 58.5 | 100.0 | 22.5 | 33.0 | 99.0 |
| Shortest path | K | 0.2 | 0.5 | 78.5 | 96.2 | 23.5 | 36.2 | 85.2 |
| | LR | 61.7 | 1.2 | 77.8 | 97.8 | 19.8 | 40.3 | 81.3 |
| | B | 0.5 | 0.5 | 79.5 | 98.0 | 17.5 | 50.3 | 83.7 |
| | HSSD | 0.0 | 3.0 | 82.0 | 100.0 | 65.5 | 32.5 | 98.0 |

*Table 18.* Question-level accuracy on completed answers for **reasoning models**.

| Question | Room | GPT-5 | gpt-oss 120b | DeepSeek R1-0528 | Gemini Flash 2.5 | GPT-5 mini-2025 | gpt-oss 20b | Qwen3 30B Think. |
|---|---|---|---|---|---|---|---|---|
| Pair distance | K | 99.8 | 98.0 | 99.5 | 99.7 | 100.0 | 94.8 | 99.7 |
| | LR | 98.8 | 99.3 | 99.8 | 100.0 | 99.8 | 94.1 | 99.8 |
| | B | 98.3 | 99.5 | 99.3 | 99.8 | 99.8 | 94.5 | 100.0 |
| | HSSD | 69.0 | 88.2 | 85.0 | 92.6 | 98.5 | 66.9 | 64.3 |
| Placement | K | 97.5 | 92.0 | 93.8 | 98.1 | 97.2 | 89.7 | 96.7 |
| | LR | 97.6 | 89.2 | 88.5 | 98.5 | 96.3 | 89.5 | 97.9 |
| | B | 95.8 | 83.6 | 78.4 | 97.7 | 93.1 | 83.4 | 97.2 |
| | HSSD | 95.9 | 85.0 | 84.0 | 100.0 | 90.4 | 87.7 | 96.6 |
| Reposi- tioning | K | 83.3 | 85.5 | 86.4 | 91.4 | 84.5 | 71.4 | 66.2 |
| | LR | 85.5 | 89.8 | 93.2 | 93.2 | 92.8 | 88.3 | 82.5 |
| | B | 78.0 | 83.3 | 86.1 | 86.6 | 84.3 | 78.7 | 75.5 |
| | HSSD | 50.5 | 60.8 | 71.4 | 77.1 | 74.8 | 51.6 | 73.0 |
| Free space | K | 83.5 | 99.0 | 98.4 | 95.2 | 99.5 | 95.6 | 99.3 |
| | LR | 47.1 | 83.5 | 33.0 | 30.3 | 79.8 | 61.6 | 0.0 |
| | B | 50.6 | 87.5 | 39.7 | 48.1 | 83.0 | 80.3 | 1.2 |
| | HSSD | 20.5 | 43.4 | 31.7 | 28.6 | 33.3 | 40.0 | 66.7 |
| Visibility | K | 94.8 | 95.6 | 97.3 | 98.8 | 98.0 | 91.8 | 97.5 |
| | LR | 95.2 | 95.6 | 97.8 | 95.8 | 98.3 | 90.2 | 96.2 |
| | B | 94.2 | 93.6 | 96.4 | 95.7 | 97.0 | 89.3 | 96.2 |
| | HSSD | 57.0 | 73.7 | 76.9 | 100.0 | 89.7 | 64.1 | 75.0 |
| View angle | K | 96.2 | 98.5 | 98.9 | 97.9 | 99.4 | 94.4 | 99.5 |
| | LR | 93.3 | 97.3 | 98.5 | 96.3 | 98.6 | 93.2 | 99.0 |
| | B | 95.2 | 98.2 | 98.7 | 97.4 | 99.4 | 92.4 | 99.2 |
| | HSSD | 59.5 | 80.0 | 87.1 | 81.6 | 94.4 | 60.5 | 76.5 |
| Max box | K | 48.5 | 62.8 | 72.3 | 84.6 | 86.9 | 44.0 | 44.6 |
| | LR | 17.3 | 28.2 | 21.9 | 0.0 | 64.0 | 9.8 | 41.7 |
| | B | 13.0 | 30.3 | 27.1 | 0.0 | 64.2 | 13.3 | 27.3 |
| | HSSD | 5.0 | 9.5 | 6.0 | 0.0 | 21.9 | 0.8 | 0.0 |
| Shortest path (valid) | K | 64.8 | 64.5 | 57.4 | 82.6 | 68.4 | 67.4 | 95.5 |
| | LR | 55.2 | 66.8 | 56.4 | 69.2 | 66.9 | 63.1 | 89.3 |
| | B | 58.5 | 58.1 | 38.2 | 50.0 | 63.0 | 60.4 | 87.8 |
| | HSSD | 28.5 | 34.5 | 47.2 | 0.0 | 47.8 | 20.0 | 50.0 |
| Shortest path (Fréchet) | K | 56.4 | 55.8 | 53.5 | 82.6 | 66.0 | 66.1 | 94.4 |
| | LR | 32.2 | 41.0 | 42.1 | 61.5 | 59.2 | 47.8 | 87.5 |
| | B | 39.5 | 45.1 | 29.3 | 41.7 | 57.8 | 48.7 | 87.8 |
| | HSSD | 22.5 | 27.3 | 13.9 | 0.0 | 34.8 | 20.7 | 50.0 |

