# OpenReview forum: "FloorplanQA: A Benchmark for Spatial Reasoning in LLMs using Structured Representations"
_ICML.cc/2026/Conference — ICML 2026 regular_

### Official Review · Reviewer_ZF6u · 2026-03-12

**Soundness:** 3
**Presentation:** 3
**Significance:** 2
**Originality:** 2
**Overall Recommendation:** 3
**Confidence:** 3

**Summary:**

This paper introduces FloorplanQA, a benchmark designed to evaluate spatial reasoning abilities of language models on structured indoor layouts represented as symbolic floorplans (e.g., JSON/XML). The benchmark includes several task categories such as distance estimation, visibility reasoning, path planning, and constrained object placement. The authors evaluate a range of frontier language models and show that current systems struggle with precise geometric reasoning over structured spatial environments.

**Compliance With Llm Reviewing Policy:**

Affirmed.

**Final Justification:**

My main concern regarding the benchmark’s necessity and distinct contribution remains unresolved. I will maintain my score and defer to the area chair for the final decision.

**Key Questions For Authors:**

1. What is the clearest scientific question that FloorplanQA enables which existing spatial reasoning benchmarks cannot cleanly evaluate?
2. Since tool use improves arithmetic-heavy tasks but not planning-heavy ones, can the benchmark more clearly separate failures caused by calculation errors from failures in spatial representation or reasoning?

**Limitations:**

The paper should discuss more explicitly the gap between symbolic 2D floorplan reasoning and richer 3D or embodied spatial reasoning settings.

**Strengths And Weaknesses:**

### Strengths
1. The paper studies an under-explored but practically relevant setting: symbolic geometric reasoning over structured indoor layouts rather than image-based or qualitative spatial tasks.
2. The benchmark covers a diverse set of spatial reasoning tasks, including metric reasoning, visibility constraints, placement feasibility, and path planning.
3. The analysis provides useful observations about model behavior, particularly that tool use substantially helps arithmetic-heavy tasks but much less for planning-style spatial reasoning.

### Weaknesses
1. As a benchmark paper, the evaluation and analysis remain somewhat limited relative to the goal of establishing a new standard benchmark; stronger validation of dataset construction, ambiguity control, and difficulty calibration would strengthen the work.
2. The novelty is somewhat narrow because several task families are natural extensions of prior spatial reasoning evaluations, and the paper does not yet fully demonstrate why this specific benchmark composition is indispensable compared to existing spatial reasoning benchmarks.

---

> ### Author Rebuttal · Authors · 2026-03-30
>
> We thank the reviewer for the constructive feedback and for recognizing the diverse set of tasks and analysis in our symbolic geometric reasoning setting.
>
> > Can the benchmark more clearly separate failures caused by calculation errors from failures in spatial representation or reasoning?"
>
> This is a valid concern. We already have evidence addressing this: Table 2 in the main text shows the tool-use results and we discuss failure cases in Appendix I, but we agree the separation between error types could be stated more explicitly. In addition to Table 2, we have Table 8, Appendix F which shows how each task responds to tool usage in more detail. For simple tasks like Pair Distance (+43 to 99%) or View Angle (+41 to 96%), failures clearly stem from calculation mistakes (e.g., wrong centroid computation) that code execution fixes entirely. For tasks like Free Space (+28 to 44%) and Repositioning (+36.5 to 83.5%), tools help with arithmetic but cannot fully resolve the spatial reasoning involved. For Max Box (−4.5 to 3%) and Shortest Path (−12.5 to 12.5%), performance actually drops — tools raise false constraint violations, produce incorrect logic, and consume more tokens from the budget.
>
> While we did not grade all reasoning logs directly, analyzing the performance change when tools handle computation, together with the failure cases, gives us insight into which problems are fundamentally about calculation vs reasoning. For instance, in Free Space a model may subtract overlapping objects individually instead of computing their union; in Max Box a model returns a suboptimal rectangle despite a larger valid one existing (Figure 8a). These are not arithmetic mistakes but conceptual ones, and they vary across questions in ways that are hard to classify automatically. We will make this inference more explicit in the revised text.
>
>  ---
>
> > "What is the clearest scientific question that FloorplanQA enables which existing spatial reasoning benchmarks cannot cleanly evaluate?"**
>
> The core question is: can LLMs reliably perform constraint-valid geometric reasoning over symbolic layout representations, as required for evaluating or validating designs? Existing spatial benchmarks either use visual inputs where perception errors dominate, or test qualitative relations ("left of", "above") without metric coordinates, making it hard to isolate reasoning failures. FloorplanQA provides fully specified symbolic layouts (coordinates, polygons) and removes perceptual uncertainty, forcing models to operate directly on geometric structure. This matters for emerging agentic workflows where LLMs are used not only to generate layouts but also to evaluate them, as well as tools that assist designers. An agent validating a design needs to answer questions like "is this placement valid given clearance constraints?" or "can a person reach the fridge?", which rely on exactly the geometric skills (placement feasibility, visibility, collision-free paths) that FloorplanQA measures. Our results show that even when arithmetic is controlled or assisted, models still fail to satisfy spatial constraints, making this an interesting and valid problem to study. While some individual question types have been studied before, we evaluate them in a new setting with diverse polygons, objects, and layout complexity.

---

> > ### Author Rebuttal · Reviewer_ZF6u · 2026-04-05
> >
> > I will maintain my current score, as my main concern regarding the benchmark’s necessity and distinct contribution remains unresolved, and I will consider other reviewers’ opinions.

---

> > > ### Author Response · Authors · 2026-04-07
> > >
> > > We thank the reviewer for the thoughtful feedback. While we understand that the concerns remain only partially resolved from the reviewer’s perspective, we believe the paper presents a strong standalone benchmark with solid empirical support, and the discussion helped us identify where its contribution should be stated more clearly.

---

### Official Review · Reviewer_rYM2 · 2026-03-13

**Soundness:** 3
**Presentation:** 3
**Significance:** 3
**Originality:** 3
**Overall Recommendation:** 4
**Confidence:** 4

**Summary:**

The paper proposed FloorPlanQA, a new benchmark designed to evaluate the LLM's ability to perform question answering based on structured input (i.e., layout in JSON or XML format). The benchmark includes 2000 layouts: a synthetic layout generated by Gemini and a realistic layout curated from HSSD-200. For question generation, the author used a template-based approach to construct an 8-taxon set to evaluate spatial understanding from low to high levels. The authors evaluate the performance of 15 LLMs, including both non-reasoning and reasoning models. The results reveal that most of the models struggle to reason over structured output, with low performance across all evaluated tasks.

**Compliance With Llm Reviewing Policy:**

Affirmed.

**Key Questions For Authors:**

1. Why is the reason for Qwen3 lower when the reasoning is enabled for the model?

2. Is there any failure in the generation of the layout provided by Gemini? Is there any quality control or review of the generated layout before it is included in the proposed benchmark?

3. How many questions are generated for each question type? Is there a way to define the difficulty or increase the number of reasoning steps for each question type?

**Limitations:**

Yes

**Strengths And Weaknesses:**

# Strengths

- The paper proposed a new benchmark for layout understanding that is publicly available.

- The synthetic portion of the benchmark is automatically generated and described sufficiently for reproducibility.

-  The authors also provide two approaches to improve the performance of the model, including tool calling and visual input, which could be explored further in the future to improve the performance in this task.

- Future direction is provided and discussed to further improve the area in the future.

- The paper is well-written and easy to follow. The figure helps clarify the task and the results.

---

# Weaknesses

- The results of the Qwen3 Thinking model seem to be lower than the non-thinking model. This seems to contradict the discussion provided in the results section. Exploring this might reveal a better understanding of the reasoning model for this task.

- The questions generated are based on a templated approach. Including a variation or a small subset of human- or LLM-generated questions might further strengthen the paper's contribution.

- While authors mentioned that HSDD is challenging because of the complexity, there is no numerical analysis of this statement. Adding the correlation between layout properties and the accuracy of this might strengthen this claim.

---

> ### Author Rebuttal · Authors · 2026-03-30
>
> We thank the reviewer for acknowledging our pipeline and paper structure, as well as for valuable feedback. Here we provide some clarifications on the questions and concerns raised above:
>
> > Why is Qwen3’s accuracy lower when reasoning is enabled?
>
> Qwen3-30B-A3B-Thinking uses 5,882 avg output tokens with 37.4% of the responses truncated due to the token budget, while Qwen3-30B-A3B-Instruct uses 3,495 avg tokens with 9.8% truncation.
>
> This can be treated as a practical limitation of models that embed the detailed reasoning into the output token stream. Increasing the budget would reduce truncation but would also increase costs proportionally.
>
> We have tables that evaluated accuracy only for samples with non-truncated outputs (Table 13, 15, App K).
>
> ---
>
> > Is there any quality control or review of the generated layout
>
> Yes, we apply systematic quality control. Approximately one-third of initially generated layouts were filtered out through automated geometric constraints, followed by manual review.
>
> Automated filtering (Appendix B.3) enforces non-overlapping bounding boxes, door clearance zones, wall-adjacency for appliances, and other geometric/functional constraints — iteratively developed to address common failure modes in Gemini-generated layouts. After automated filtering, a visual inspection pass was performed, to identify issues not captured programmatically, such as enclosed kitchens without access. Common rejected cases included objects blocking doors/passages, appliances (fridge, stove, etc.) placed in the middle of the room without functional justification, and wall-object collisions.
>
> ---
>
> > Templated questions, difficulty, and HSSD complexity
>
> **Question generation.** We generate 1 question per question type per layout (8 questions × 2,000 layouts = 16,000 pairs). Objects referenced in each question are selected randomly from the layout. While questions follow templates, the spatial reasoning required is unique to each layout — the same template applied to different rooms produces different computational problems (different object counts, polygon shapes, spatial configurations).
>
> **Numerical analysis of layout complexity.** We computed Spearman correlations between layout properties and per-layout accuracy (averaged across all 15 models, 7 question types, 2,000 layouts). The strongest predictors of difficulty are $N_{\text{overlaps}}$ (r=−0.517, p<1e-137), total polygon vertices (r=−0.451, p<1e-101), and area (r=−0.303, p<1e-44). $N_{\text{objects}}$  also correlates negatively (r=−0.269, p<1e-34). For shortest_path, HSSD paths require more waypoints (mean 3.6 vs 2.1–2.8), with accuracy dropping accordingly (16.6% vs 36–42%). We plan to include a detailed analysis in the Appendix.
>
> **Controlling difficulty.** Difficulty can be controlled by selecting layouts with more objects, overlaps, or complex polygons. The per-question-type correlations shows that difficulty can be adjusted through layout complexity, primarily affecting global tasks (free_space, max_box) rather than local ones (pair_distance, view_angle).

---

> > ### Author Rebuttal · Reviewer_rYM2 · 2026-04-03
> >
> > Thanks for the detailed response. Please add the new results and details to the paper later.
> > I would like to maintain my positive review score.

---

### Official Review · Reviewer_LNwQ · 2026-03-13

**Soundness:** 3
**Presentation:** 3
**Significance:** 2
**Originality:** 3
**Overall Recommendation:** 4
**Confidence:** 4

**Summary:**

FloorplanQA is a benchmark for testing LLMs’ spatial reasoning on indoor floor plans using structured, symbolic layouts (JSON/polygons with rooms, doors/windows, objects, and sizes) rather than images or external tools. It targets three capability groups—metric (e.g., distances, areas), topological (e.g., visibility, occupancy, placeability), and action/path reasoning (e.g., relocation, shortest paths with safety margins). The dataset contains 2000 layouts (1800 synthetically generated and 200 derived from HSSD), with 8 questions per layout for a total of 16000 QA pairs. A unified automatic scoring protocol is provided, with explicit accounting for invalid outputs caused by formatting or truncation, enabling fair comparisons and detailed error diagnosis across models.

**Compliance With Llm Reviewing Policy:**

Affirmed.

**Ethical Review Flag:**

Flag this paper for an ethics review.

**Final Justification:**

Compared to the initial version of the paper, the rebuttal has answered my questions. So I raise my rating to 4.

**Key Questions For Authors:**

With only symbolic floor plans, this isn’t directly usable. Could you also provide corresponding image-based modeling (e.g., via image-generation models) and test the QA with the SOTA VLM model？

And can you report the token cost for each model? I think this is also an important part for the evaluation.

If author can provide more results with VLM, I will consider raising my rating.

**Limitations:**

yes

**Strengths And Weaknesses:**

Strength:
1. Symbolic input, tool-free setup that isolates pure geometric/topological reasoning without visual noise or help from external solvers.
2. Comprehensive coverage across metric, topological, and action/path tasks, including Free Space, Max Box, Placement, Visibility, and Shortest Path.
3. Strong comparability via automation: strict output formats and tolerance thresholds; tailored scoring rules for numbers, sets, and sequences (e.g., 2–5% tolerances, set matching, Fréchet threshold with collision constraints).
4. Compare to the ICLR version, the paper extend the 4.2 and discuss the experiment results on the object reference substitution and the semantic perturbation.

Weakness:
1. Planar-geometry focus: limited coverage of richer functional metrics (e.g., door flow, dynamic crowds, reachability and behavior constraints). In detail, to precept the real-world, only 2D floorplan has the limitation.
2. Sensitive to long-context/token budgets: truncation and formatting issues materially affect outcomes.
Compare to the ICLR version, authors' shift to the Douglas-Peucker algorithm in the latest version precisely highlights the severe bottleneck of token budgets when dealing with complex real-world layouts like HSSD.
3. Although the task is framed as "Floorplan QA," the paper primarily evaluates text-only LLMs using semantic descriptions encoded in JSON format. While the authors briefly test the vision capabilities of GPT-4.1 in the appendix, the conclusions drawn from these multimodal experiments are highly debatable. Given the rapid advancement in multimodal capabilities, the authors should evaluate more recent, SOTA VLMs (e.g., Gemini 3.1) to provide a more convincing and up-to-date assessment.

---

> ### Author Rebuttal · Authors · 2026-03-29
>
> We thank the reviewer for these valuable comments and raised questions.
>
> > Image modeling and VLM tests
>
> We've conducted extra vision experiments to evaluate how modern VLMs perform on FloorplanQA when given rendered floorplan images. The original paper included VLM experiments with both labeled and icon-based images, but only in combination with JSON layouts. Here, we extend this by additionally testing image-only conditions (no text at all).
>
> ## Setup
> We evaluate **3 VLM models** across **5 input conditions** on all 8 question types (12,800 requests per model):
>
> |Model|Max Tokens|
> |-|-|
> |Gemini 3.1 Flash Lite|8,192|
> |GPT-5-mini|8,192|
> |GPT-4.1-mini|8,192|
>
> We chose "mini" variants as they achieve near-parity with full-size options while being cheaper and faster. Claude models were excluded due to higher API pricing.
>
> **Image Types:**
> - **Labeled floorplan**: Top-down rendering (Fig 4, right) with text labels for each object, and a metric coordinate grid (X/Y axes in meters).
> - **Icon floorplan**: Top-down rendering with realistic furniture icons (Fig 2) and a metric coordinate grid, but *no text labels*.
>
> **Input Conditions:**
> |Condition|Image|Description|
> |-|-|-|
> |JSON-only|None|Structured text baseline|
> |JSON+Labeled|Labeled|Structured text + annotated image|
> |JSON+Icon|Icon|Structured text + icon image|
> |Labeled-only|Labeled|Image with boxes and text labels|
> |Icon-only|Icon|Icons only, no text — purely visual|
>
> **Datasets:** HSSD (200 real-world rooms) and Mixed (200 synthetic: 75 living, 75 bed, 50 kitchen). Icon conditions apply only to Mixed.
> ## Results
> ### Table 1: Avg Accuracy by Condition (%, HSSD+Mixed)
>
> |Model|JSON-only|JSON+Labeled|Labeled-only|
> |-|-|-|-|
> |GPT-4.1-mini|47.38|45.50|21.22|
> |GPT-5-mini|54.44|60.78|27.09|
> |Gemini3.1FL|55.22|56.78|35.03|
>
> ## Table 2: Accuracy by Condition and Room Type (%)
>
> |Model|Room|JSON-only|JSON+Labeled|JSON+Icon|Labeled-only|Icon-only|
> |-|-|-|-|-|-|-|
> |GPT-4.1-mini|HSSD|35.44|34.81|—|19.38|—|
> |GPT-4.1-mini|Mixed|59.31|56.19|55.88|23.06|18.94|
> |GPT-5-mini|HSSD|31.75|37.75|—|23.12|—|
> |GPT-5-mini|Mixed|77.12|83.81|83.81|31.06|20.06|
> |Gemini3.1FL|HSSD|43.50|45.19|—|30.25|—|
> |Gemini3.1FL|Mixed|66.94|68.38|64.00|39.81|23.25|
>
> ### Table 3: Token Usage and Cost
>
> |Model|AvgOut|AvgIn|Requests|Cost, $|
> |-|-|-|-|-|
> |GPT-4.1-mini|1,269|2,279|12,800|18.83|
> |GPT-5-mini|4,283|2,133|12,800|57.90|
> |Gemini3.1FL|2,734|3,482|12,800|17.30|
>
> *Pricing: GPT-4.1-mini batch (`$0.20/$0.80` per 1M in/out tokens), GPT-5-mini batch (`$0.125/$1.00`). Gemini cost from actual billing. Total experiment cost: ~$94.03.*
>
> ### Key Findings
> 1. **Images + JSON: modest gains.** JSON + Labeled images improves over JSON-only by +1.5–6.3pp, with the largest gains on free_space, max_box, and shortest_path. GPT-5-mini is the main beneficiary (+6.34pp), while GPT-4.1-mini shows no benefit (−1.88pp), suggesting some models struggle to operate with unusual formats, but spend more tokens processing them.
>
> 2. **Image-only drops heavily (−20–30pp).** Labeled-only reaches 19–40% across models and datasets, and on Mixed, Icon-only drops further to 18–23%. Models must read text labels from images, estimate coordinates from the grid, and do arithmetic over noisy visual estimates — each step compounds error. Icon-only is even harder: furniture icons can be visually ambiguous (e.g., a nightstand vs. a small table) and vary in style.
>
> We will add new results to the main paper.
>
> ---
>
> > Token cost
>
> Here we report token usage for all 15 models on the full benchmark. That table and cost breakdown (which require price adjustments on the day of the experiments) will be included in the Appendix.
>
> ## Table 4: Avg Token Usage per Model
>
> |Model|AvgIn|AvgOut|AvgReas|Total, M|
> |-|-|-|-|-|
> |GPT-5|1,876|1,056|293|45.7|
> |gpt-oss-120B|1,934|2,743|—|77.6|
> |DeepSeek-R1-0528|1,582|7,473|—|144.9|
> |GPT-5-mini|1,873|4,319|3,452|99.1|
> |Gemini2.5Flash|2,037|1,342|8,323|44.5|
> |gpt-oss-20B|1,938|3,860|—|92.8|
> |Qwen3-30B-Think|2,052|5,882|—|127.0|
> |ClaudeSonnet4|1,947|866|—|45.0|
> |GPT-4.1|1,874|1,367|—|51.9|
> |Kimi-K2|1,873|685|—|40.9|
> |Qwen3-Coder-480B|2,050|1,328|—|54.0|
> |Qwen3-235B|1,939|4,682|—|105.9|
> |GPT-4.1-mini|1,874|1,583|—|55.3|
> |Qwen3-30B|2,048|3,495|—|88.7|
> |Devstral-Small|2,073|1,101|—|50.8|
>
> *Reasoning tokens reported where available (GPT-5, Gemini). For DeepSeek-R1, Qwen3-Think, reasoning is included in the output.*
>
> ---
>
> > Sensitive to long-context/token budgets
>
> Truncation can affect verbose reasoning models, but token limits and associated costs can also be considered as constraints. Formatting issues during parsing led to fluctuations of <1% in accuracy (Table 7, App E). The Douglas–Peucker algorithm was introduced not only to reduce token usage by simplifying polygon representations, but also to clarify the task and make it more realistic. Since floorplans are derived from 3D scenes, fine details such as table legs are not essential for floor plan representation (Fig 4, App C).

---

> > ### Author Rebuttal · Reviewer_LNwQ · 2026-04-03
> >
> > Thanks for authors reply. I really appreciate more experiments with the VLM. But I still do not see the rendering image of the floorplan with the image generation model. I hope authors can do the experiment after rendering the floorplan, but not only work on the simple floorplan.

---

> > > ### Author Response · Authors · 2026-04-06
> > >
> > > Thank you for the follow-up. We conducted an additional experiment: generating realistic top-down floorplan images with an image generation model and evaluating VLMs on the full benchmark using these rendered images.
> > >
> > > **Image Generation.** We used Nano Banana 2 (Gemini 3.1 Flash Image Preview) in image-to-image mode: each schematic floorplan render was transformed into a realistic top-down visualization while mainly preserving furniture positions and sizes. That format was easier for an image model to render than a professional architectural blueprint. We also tested Qwen-Image-2.0, but it failed to maintain spatial proportions and positions (see examples in the dataset link below), so it was excluded from the benchmark. All 200 rooms from the Mixed dataset were generated. Examples of all rendering styles (simple bounding boxes, icons, Nano Banana 2, and Qwen) are available at: https://kaggle.com/datasets/88ae0efda62985f0089ae992c7e0fd7d82f9868fd2d0dc2936266638cdf7b315 (the link is anonymous in accordance with 7. Additional details regarding the discussion period, Rebuttal instructions for authors).
> > >
> > > **Experimental Setup.** We tested the same two conditions with the AI-generated images:
> > > - **JSON+GenImg**: AI-generated realistic image + JSON layout (same structured data as the original benchmark)
> > > - **GenImg Only**: AI-generated realistic image alone, with room dimensions provided as scale reference
> > >
> > > We evaluated the same 3 VLMs (GPT-4.1-mini, GPT-5-mini, Gemini 3.1 Flash Lite) across all 8 question types on the Mixed dataset (200 rooms).
> > >
> > > **New columns for Table 2 (Mixed dataset, accuracy %):**
> > >
> > > | Model | JSON+GenImg | GenImg-only |
> > > |---|---|---|
> > > | GPT-4.1-mini | **55.69** | **19.12** |
> > > | GPT-5-mini | **81.75** | **24.31** |
> > > | Gemini 3.1 FL | **63.81** | **25.00** |
> > >
> > > **Per-question-type breakdown (Mixed dataset, accuracy %):**
> > >
> > > | Model | Condition | Pair Dist. | Free Space | Max Box | View Angle | Obstruction | Placement | Repos. | Short. Path | **Avg** |
> > > |---|---|---|---|---|---|---|---|---|---|---|
> > > | GPT-4.1-mini | Icon+JSON | 97.0 | 17.0 | 7.5 | 92.5 | 70.5 | 71.5 | 47.5 | 43.5 | 55.88 |
> > > | GPT-4.1-mini | GenImg+JSON | 98.0 | 17.5 | 8.5 | 90.5 | 71.5 | 77.0 | 40.0 | 42.5 | **55.69** |
> > > | GPT-4.1-mini | Icon-only | 10.0 | 4.5 | 4.5 | 12.5 | 11.0 | 82.0 | 6.0 | 21.0 | 18.94 |
> > > | GPT-4.1-mini | GenImg-only | 5.0 | 7.0 | 6.0 | 15.5 | 17.5 | 74.5 | 5.0 | 22.5 | **19.12** |
> > > | | | | | | | | | | | |
> > > | GPT-5-mini | Icon+JSON | 99.5 | 93.0 | 61.5 | 92.0 | 97.0 | 92.0 | 72.0 | 63.5 | 83.81 |
> > > | GPT-5-mini | GenImg+JSON | 99.0 | 90.5 | 69.0 | 93.5 | 97.0 | 91.5 | 61.5 | 52.0 | **81.75** |
> > > | GPT-5-mini | Icon-only | 7.5 | 5.5 | 5.5 | 11.5 | 17.0 | 86.0 | 5.0 | 22.5 | 20.06 |
> > > | GPT-5-mini | GenImg-only | 8.0 | 12.0 | 3.5 | 14.5 | 31.5 | 84.5 | 7.0 | 33.5 | **24.31** |
> > > | | | | | | | | | | | |
> > > | Gemini 3.1 FL | Icon+JSON | 100.0 | 36.0 | 17.0 | 83.0 | 73.0 | 85.5 | 62.5 | 55.0 | 64.00 |
> > > | Gemini 3.1 FL | GenImg+JSON | 100.0 | 33.0 | 16.0 | 87.0 | 69.5 | 85.0 | 64.0 | 56.0 | **63.81** |
> > > | Gemini 3.1 FL | Icon-only | 12.5 | 9.0 | 8.0 | 17.5 | 23.0 | 82.5 | 10.5 | 23.0 | 23.25 |
> > > | Gemini 3.1 FL | GenImg-only | 11.0 | 6.5 | 6.0 | 17.5 | 29.5 | 82.0 | 12.0 | 35.5 | **25.00** |
> > >
> > > **Findings:**
> > >
> > > 1. **Replacing schematic icons with AI-generated realistic images has no meaningful effect.** JSON+GenImg and JSON+Icon produce nearly identical results across all models and question types (avg difference <2 pp). Similarly, GenImg-only and Icon-only are in the same accuracy range (18.94-23.25 vs 19.12-25.00).
> > >
> > > 2. **Per-question trends.** In the image-only setting, GenImg slightly outperforms Icons on obstruction detection (17.5-29.5 vs 11.0-23.0) and shortest path (22.5-35.5 vs 21.0-23.0), likely because realistic renderings make furniture shapes easier to recognize. Placement (yes/no) stays stable at ~75-86% in both conditions since it needs less metric precision. In the JSON+image setting, GenImg+JSON shows a small boost on max box for GPT-5-mini (69.0 vs 61.5) and on view angle for Gemini (87.0 vs 83.0), suggesting realistic images can occasionally help models better interpret object shapes and orientations. Repositioning drops slightly with GenImg for GPT-5-mini (61.5 vs 72.0), possibly because spatial distortion in generated images confuses movement direction reasoning. Overall, though, per-question differences between GenImg and Icons stay within a few pp when JSON is present, making the help of different visualization styles not really significant.
> > >
> > > We will release the generated images and the generation pipeline alongside the benchmark data. To improve VLM performance on floorplan images, a natural next step would be to collect paired data: real floorplan photographs matched with ground-truth spatial layouts, and then fine-tune on that. Current VLMs struggle to reason over different formats of flooplans. We see this as a promising follow-up direction.

---

### Decision · Program_Chairs · 2026-04-30

**Decision:**

Accept (regular)

**Comment:**

This paper introduces a benchmark for evaluating LLMs on spatial reasoning over structured symbolic indoor layouts. The core idea is to isolate pure geometric and topological reasoning by providing models with JSON or XML encoded floorplans rather than images, removing perceptual noise and forcing models to operate directly on spatial structure. The benchmark covers a diverse set of tasks including distance estimation, visibility, constrained placement, and path planning, and evaluates frontier LLMs showing that most models struggle significantly, especially on tasks requiring constraint satisfaction and multi-step spatial reasoning. Overall this is a solid benchmark contribution.